# Golgi-independent secretory trafficking through recycling endosomes in neuronal dendrites and spines

Aaron B Bowen[1], Ashley M Bourke[1], Brian G Hiester[1], Cyril Hanus[2], Matthew J Kennedy[1]*

[1]Department of Pharmacology, University of Colorado School of Medicine, Aurora, United States; [2]Center for Psychiatry and Neurosciences, University Paris-Descartes, Paris, France

**Abstract** Neurons face the challenge of regulating the abundance, distribution and repertoire of integral membrane proteins within their immense, architecturally complex dendritic arbors. While the endoplasmic reticulum (ER) supports dendritic translation, most dendrites lack the Golgi apparatus (GA), an essential organelle for conventional secretory trafficking. Thus, whether secretory cargo is locally trafficked in dendrites through a non-canonical pathway remains a fundamental question. Here we define the dendritic trafficking itinerary for key synaptic molecules in rat cortical neurons. Following ER exit, the AMPA-type glutamate receptor GluA1 and neuroligin 1 undergo spatially restricted entry into the dendritic secretory pathway and accumulate in recycling endosomes (REs) located in dendrites and spines before reaching the plasma membrane. Surprisingly, GluA1 surface delivery occurred even when GA function was disrupted. Thus, in addition to their canonical role in protein recycling, REs also mediate forward secretory trafficking in neuronal dendrites and spines through a specialized GA-independent trafficking network.
DOI: https://doi.org/10.7554/eLife.27362.001

*For correspondence:
matthew.kennedy@ucdenver.edu

**Competing interests:** The authors declare that no competing interests exist.

## Introduction

Neurons face the challenge of tuning protein levels in dendritic processes that can project hundreds of micrometers from their cell bodies. Rather than solely relying on diffusion or long-range active transport of proteins synthesized in the cell body (*Williams et al., 2016*), many dendritic proteins are synthesized near their site of action. Indeed, mRNAs encoding thousands of different proteins are found in dendrites (*Cajigas et al., 2012*), where their regulated translation is critical for neuronal dendrite development, maintenance and plasticity (*Sutton and Schuman, 2006*; *Bramham and Wells, 2007*; *Hanus and Schuman, 2013*). Intriguingly, many of these dendritic mRNAs encode secreted factors and critical integral membrane proteins, implying the presence of a satellite secretory system for local processing and trafficking of newly synthesized cargo. In addition to locally translated proteins, nascent proteins synthesized in the soma can also enter dendritic compartments through lateral diffusion in the ER membrane (*Cui-Wang et al., 2012*), or by active transport (*Valenzuela et al., 2014*), since the ER network is contiguous throughout the somatodendritic domain. In contrast, most dendrites (~80%) lack Golgi apparatus (GA), a defining organelle of the cellular secretory network that mediates canonical post-ER trafficking (*Krijnse-Locker et al., 1995*; *Torre and Steward, 1996*; *Gardiol et al., 1999*; *Hanus et al., 2016*). Thus, whether dendrites can support local secretory trafficking in the absence of GA and if so, the identity and spatial distribution of the organelles responsible have remained fundamental issues.

Previous studies demonstrate that cargo exiting the dendritic ER is initially directed to ER-Golgi intermediate compartments (ERGICs), a collection of tubulovesicular membranes distributed

**eLife digest** All cells must produce, sort and deliver molecular building blocks to the right places at the right time and in appropriate amounts. This is particularly important for neurons, which are the largest and most structurally complex cells in the body. A typical neuron consists of a cell body covered in branches called dendrites, plus a single cable-like structure known as an axon. Dendrites receive inputs from other neurons and relay the information to the cell body in the form of electrical signals. The cell body processes these electrical signals and the resulting signals then travel along the axon to terminals at the far-end. The axon terminals in turn pass the signals on to the dendrites of other neurons via junctions called synapses.

For synapses to work correctly, the membranes surrounding the dendrites need to contain receptor proteins that can detect incoming signals. These proteins must be continually replenished, raising the question of how newly made receptor molecules are shuttled to the appropriate locations within the dendrites.

A series of compartments called the Golgi complex play an important role in processing newly-made proteins in many different types of cells. As proteins pass through the Golgi, enzymes within the tunnel walls modify the proteins by adding or removing molecular groups. Therefore, it has been suggested that the route that the synapse receptor proteins take through the neuron to reach the dendrites always includes a visit to the Golgi. However, the Golgi complex in neurons is mostly confined to the cell body, raising the question of whether proteins that are locally produced within dendrites can make the journey to nearby synapses without visiting the Golgi complex.

Bowen et al. used a microscope to follow the movements of synapse receptor proteins through neurons grown in a dish. The experiments show that proteins destined for the dendrites make a number of stops after leaving the cell body. However, some synaptic proteins reach the dendrites without passing through the Golgi at all, suggesting neurons are much less dependent on the Golgi to process newly-made proteins than other types of cells.

Genetic mutations that prevent proteins from finding their way to their required destinations, or that disrupt the work of enzymes inside trafficking stations like the Golgi, cause numerous human diseases. Understanding how proteins travel to specific destinations inside healthy cells should also help reveal what happens when this process fails.

DOI: https://doi.org/10.7554/eLife.27362.002

throughout all dendrites (*Krijnse-Locker et al., 1995*; *Hanus et al., 2014*). The ERGIC network normally shuttles proteins between the ER and GA. However, given the lack of dendritic GA, the fate of dendritic secretory cargo following ERGIC accumulation remains unclear. In addition to mediating bi-directional trafficking at the ER-Golgi boundary (*Ben-Tekaya et al., 2005*), ERGIC membranes are also reported to physically interact with recycling endosomes (REs) (*Marie et al., 2009*). A number of secretory proteins traffic through REs on their initial voyage to the PM, and some cargoes, such as the cystic fibrosis transmembrane conductance receptor (CFTR), may enter the RE network in a GA-independent manner (*Yoo et al., 2002*; *Marie et al., 2009*; *Grieve and Rabouille, 2011*). The broad dendritic distribution of REs, including within a large fraction of dendritic spines, makes them well suited to mediate forward trafficking to the plasma membrane with high spatial precision, perhaps at the level of individual synapses (*Cooney et al., 2002*; *Park et al., 2004*; *Kennedy et al., 2010*). However, whether REs play any role in forward trafficking in neuronal dendrites remains unknown.

To better understand the organization of the neuronal secretory pathway, we employed an ER sequestration/inducible release strategy to visualize the trafficking itinerary for the AMPA-type glutamate receptor (GluA1) and the postsynaptic cell adhesion molecule neuroligin 1 (NL1) in neuronal dendrites. Following ER-release, we observed rapid, spatially confined delivery of GluA1 to dendritic ERGICs. This was immediately followed by robust accumulation in spine and dendritic REs, which were often observed in close physical proximity to ERGIC. Forward trafficking through REs to the plasma membrane still occurred even after GA function was disrupted with brefeldin A (BFA), indicating that a significant fraction of cargo enters the RE network in a GA-independent manner. Accordingly, disrupting RE function impaired surface delivery of GluA1 released from the ER.

Combined, these data support a novel GA-independent mode of local trafficking in dendrites via a satellite secretory network defined by the ERGIC and RE networks.

## Results

### Inducible release of synaptic cargo molecules from the ER

In order to specifically visualize synaptic proteins trafficking through the neuronal secretory pathway, we employed a previously described inducible ER release system based on a self-associating mutant of FKBP ($F_M$) (*Rivera et al., 2000*; *Al-Bassam et al., 2012*). Proteins of interest are trapped in the ER by fusing them to tandem repeats of $F_M$, but can be conditionally released by adding a biologically inert rapamycin analog (dimer-dimer solubilizer or DDS), which dissociates $F_M$ multimers, allowing ER exit and progression through the secretory pathway (*Rivera et al., 2000*). Thus, using live cell fluorescence microscopy, we can directly visualize a bolus of secretory cargo as it exits the ER and populates each organelle of the secretory pathway. We engineered ER-retained versions of the AMPA-type glutamate receptor GluA1 ($3xF_M$-GluA1) and the synaptic cell adhesion molecule neuroligin 1 ($4xF_M$-NL1) (*Figure 1A*). We included a consensus cleavage site for the GA-resident protease furin so that the $F_M$ repeat domains are removed as cargo transits the GA (*Figure 1A*). We initially verified inducible ER release of NL1 ($4xF_M$-NL1) in COS7 cells. Prior to addition of DDS, $4xF_M$-NL1 strongly colocalizes with an engineered ER marker (TfR-KDEL) (*Figure 1—figure supplement 1*). Following addition of 1 μM DDS, $4xF_M$-NL1 rapidly redistributes to the perinuclear GA and ultimately to the cell surface (*Figure 1—figure supplement 2*). Because AMPA receptors do not robustly traffic to the surface of non-neuronal cells, we tested ER retention and release of $3xF_M$-GluA1 in dissociated cortical neuronal cultures. Without DDS, the distribution of $3xF_M$-GluA1 closely resembled the endogenous ER-marker BIP (*Figure 1B*). Adding DDS triggered robust accumulation of $3xF_M$-GluA1 in the GA (*Figure 1B*). To verify subsequent PM delivery, we surface labeled live cortical neurons expressing $3xF_M$-GluA1 at various time-points following DDS addition using an antibody directed toward an engineered hemagglutinin (HA) tag on the extracellular domain of $3xF_M$-GluA1 (*Figure 1C*). We observed very low background levels of $3xF_M$-GluA1 on the cell surface prior to release and robust surface delivery following ER-release (*Figure 1C,D*). $4xF_M$-SEP-NL1 was also efficiently delivered to the cell surface of neurons following addition of DDS (*Figure 1E*). Significant surface delivery was detectable within two hours following ER-release for both cargoes. Maximum levels were reached at four and eight hours for NL1 and GluA1 respectively (*Figure 1D–E*). Following delivery to the PM, both GluA1 and NL1 localized to dendrites and spines, consistent with previous reports using similarly tagged expression constructs and their associated fluorescent proteins were sensitive to thrombin treatment (*Figure 1F*, *Figure 1—figure supplements 2* and *3*) (*Chih et al., 2006*; *Patterson et al., 2010*).

### Synaptic cargo molecules enter local secretory organelles in dendrites following ER exit

We first asked whether the localization of proteins leaving the ER in specific cellular domains (e.g. dendrites vs. soma) is preserved as cargo migrates to post-ER trafficking organelles. We fused the photoconvertible fluorescent protein mEOS3.2 to $3xF_M$-GluA1 and locally photoconverted ER-localized $3xF_M$-mEOS-GluA1 from green to red in either dendrites or the soma (*Zhang et al., 2012*; *Chen et al., 2013*). For dendritic release experiments, we photoconverted $3xF_M$-mEOS-GluA1 within a 30–40 μm stretch of dendritic ER located approximately 50 μm from the soma. Locally photoconverted $3xF_M$-mEOS-GluA1 (hereafter referred to as mEOS*-GluA1) displayed limited mobility in the absence of DDS, consistent with effective clustering and ER retention by the $F_M$ domains (*Figure 2—figure supplement 1*). In contrast, when we added DDS, mEOS-GluA1* exited the ER and accumulated in nearby trafficking organelles (*Video 1*). Two hours after dendritic photoconversion and ER-release, we observed that most mEOS*-GluA1 redistributed to trafficking organelles within dendrites, with only 12.9 ± 3.0% (mean ± SEM, n = 9 neurons from 3 experiments) of the photoconverted signal returning to the neuronal soma (*Figure 2A,C*). Similarly, when we locally photoconverted mEOS-GluA1 in the soma, the vast majority (93.3 ± 3.3%; mean ± SEM; n = 7 neurons from 3 experiments) of mEOS*-GluA1 redistributed to the somatic GA following ER release with negligible signal reaching even proximal dendritic regions (*Figure 2B–C*; *Video 2*). Thus, AMPA

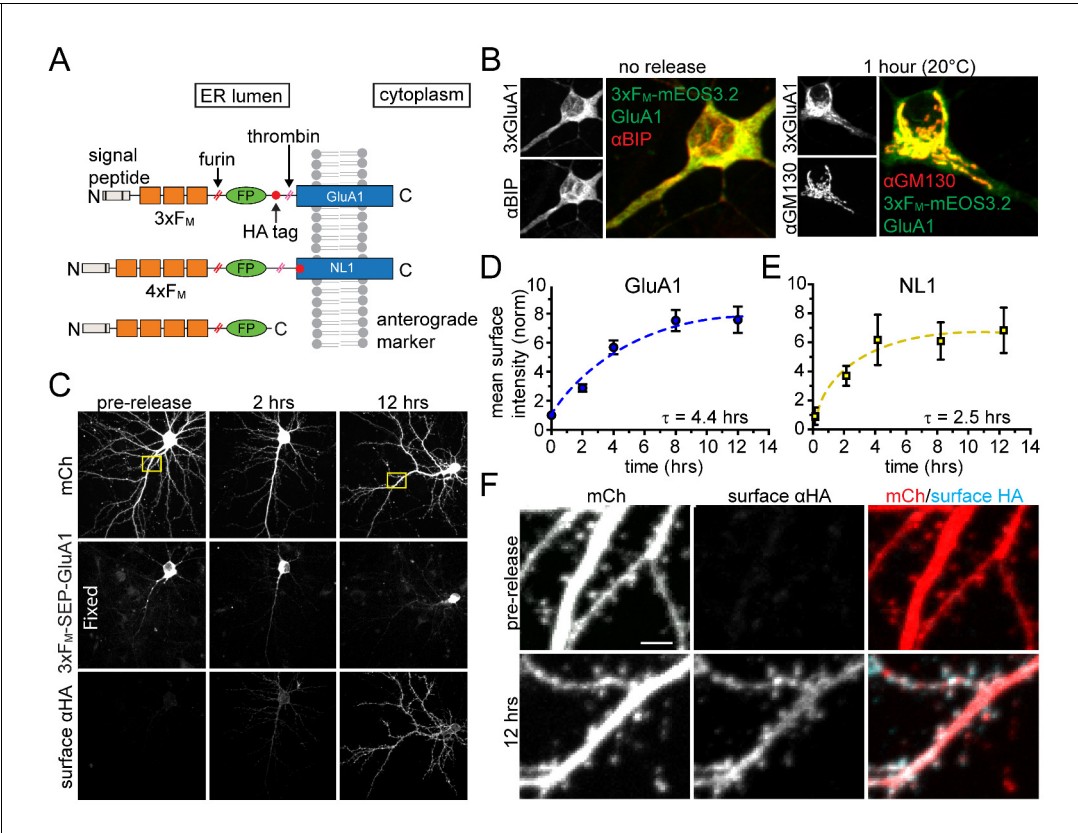

**Figure 1.** Inducible release system to investigate the dendritic secretory network. (**A**) Schematic depicting inducible release constructs. Multiple copies of self-associating $F_M$ domains were fused to target proteins downstream of an ER signal peptide and upstream of a fluorescent protein (FP). DDS dissociates $F_M$ domains allowing synchronous ER exit. A furin cleavage site allows removal of the $F_M$ domains as they transit the GA. A thrombin cleavage site was included in some constructs so that the FP could be selectively removed from proteins localized at the PM. (**B**) Comparison of $3xF_M$-mEOS-GluA1 with the endogenous ER-marker BiP before release (left panel) and the Golgi marker GM130 1 hr after DDS addition (right panel). (**C**) Detection of GluA1 surface delivery at various time points following addition of DDS by surface labelling against the extracellular HA-tag of $3xF_M$-GluA1. (**D**) Quantification of GluA1 surface delivery shown in C (mean ± SEM, n = 10–12 neurons/timepoint from 2 independent experiments). All values normalized to neurons that were not treated with DDS. (**E**) Time-course of NL1 surface delivery (mean ± SEM, n = 9–10 neurons/timepoint from 2 independent experiments). (**F**) Localization of surface GluA1 after ER-release. Images taken from insets in panel C. Scale bar, 2 µm.
DOI: https://doi.org/10.7554/eLife.27362.003

The following figure supplements are available for figure 1:

**Figure supplement 1.** ER-retention of $4xF_M$-SEP-NL1.
DOI: https://doi.org/10.7554/eLife.27362.004

**Figure supplement 2.** $4xF_M$-NL1 trafficking in COS7 cell.
DOI: https://doi.org/10.7554/eLife.27362.005

**Figure supplement 3.** Kinetics of fluorescent tag removal by thrombin.
DOI: https://doi.org/10.7554/eLife.27362.006

receptors enter distinct downstream trafficking organelles depending on the cellular domain from which they exit the ER. Interestingly, cargo leaving the dendritic ER accumulated in dendritic puncta with kinetics that closely matched the accumulation of GluA1 signal in the somatic GA in the same cells (*Figure 3A*).

To determine the spatial dynamics of GluA1 as it accumulated in organelles downstream from the ER, we quantified the mEOS\*-GluA1 content of these structures as a function of distance from the edge of photoconversion, 60 min following ER release. This analysis revealed that mEOS\*-GluA1 preferentially accumulated at organelles close to or within the initial photoconverted region with a length constant (i.e. distance over which the photoconverted signal decreases to 63.2% relative to the edge of the photoconverted region) of 11.6 µm (*Figure 3B,C*; n = 5 neurons from 3

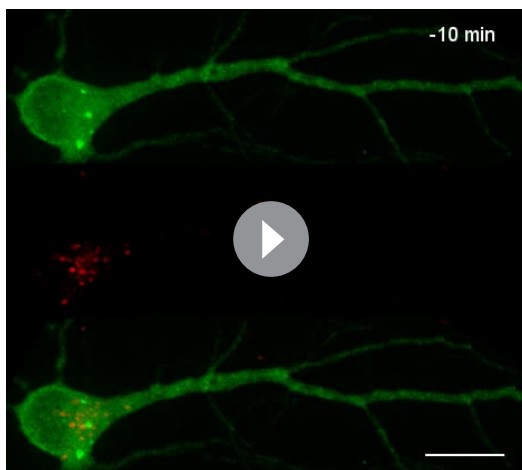

**Video 1.** Spatially restricted trafficking of mEOS-GluA1 photoconverted in the dendritic ER. Shown is a cultured cortical neuron expressing 3xF$_M$-mEOS3.2-GluA1. Left panel shows total GluA1 (green channel), middle panel shows photoconverted GluA1 (red channel) and right panel shows the merge. Focal photoconversion targeted to a segment of dendrite occurred between frames 1 and 2. DDS was added immediately after frame 2. Z-stacks were then acquired every 10 min. Scale bar, 20 μm.

DOI: https://doi.org/10.7554/eLife.27362.010

experiments). Conversely, after somatic photo-conversion, we observed minimal entry of mEOS*-GluA1 into dendritic puncta (*Figure 3D, E*). Together, these experiments reveal a striking compartmentalization of the early secretory pathway between dendrites and soma, with little crosstalk, even though the ER is contiguous between these cellular domains (*Cui-Wang et al., 2012*).

## Synaptic cargo accumulates in dendritic ERGIC following ER exit

We next investigated the identity of the dendritic organelle that first receives GluA1 following ER exit. We first tested whether cargo flux through these organelles, like GA, was sensitive to temperature. We performed temperature block experiments where we maintained cells at 20 °C during cargo release from the ER. While cargo can exit the ER at this temperature, it stalls in the GA (*Matlin and Simons, 1983*). Indeed, we observed robust accumulation of newly released GluA1 in the somatic GA under these conditions (*Figure 4A*, *Figure 4—figure supplement 1*). However, we also observed cargo stalling in punctate dendritic organelles in nearly every dendritic branch (*Figure 4B*, *Figure 4—figure supplement 1*). Based on this temperature block experiment, we initially hypothesized these structures were previously described Golgi outposts (*Horton and Ehlers, 2003*; *Horton et al., 2005*). However using antibodies against canonical GA markers we could only detect GA in a small minority of dendritic branches, usually proximal to the soma, in agreement with other studies (*Krijnse-Locker et al., 1995*; *Torre and Steward, 1996*; *Gardiol et al., 1999*). Accordingly, the dendritic GluA1 puncta that arose following ER release did not colocalize with classical *cis-* (GM130) or *trans-*GA markers (TGN38) even though the somatic GA was strongly labeled for both markers (*Figure 4A,B*, *Figure 4—figure supplement 1*). A previous study reported accumulation of a non-neuronal cargo in dendritic ERGICs shortly following ER release (*Hanus et al., 2014*). In agreement with this study, we observed ERGIC membranes throughout the dendritic arbor (*Figure 4—figure supplements 2* and *3*). We also observed that the 3xF$_M$-mCh-GluA1 puncta that formed early following ER release strongly colocalized with ERGIC53, detected either by antibody staining for endogenous p58 (rat homologue of ERGIC53) or by expressing GFP-ERGIC53 (*Figure 4C*, *Figure 4—figure supplements 2* and *3*). Colocalization between GFP-ERGIC53 and 3xF$_M$-mCh-GluA1 peaked ~50–60 min following ER-release, and declined as cargo progressed through the secretory network (*Figure 4D*). These experiments provide evidence for a local dendritic trafficking network, but we also wanted to determine whether long-range trafficking from the somatic GA to dendritic domains also occurs. To address this issue, we released 3xF$_M$-mCh-GluA1 from the ER and allowed it to accumulate in the somatic GA and dendritic ERGICs for 1 hr. We then photobleached all of the detectable dendritic signal while preserving the somatic GA signal and performed rapid timelapse imaging ~1 hr later (*Figure 4—figure supplement 4*). We observed both mobile and stationary mCh-GluA1 puncta accumulate in dendrites with a proximal (more abundant) to distal gradient during the recovery period. Thus, transport from the somatic GA to the dendritic arbor (especially proximal regions) can also occur. (*Figure 4—figure supplement 4*).

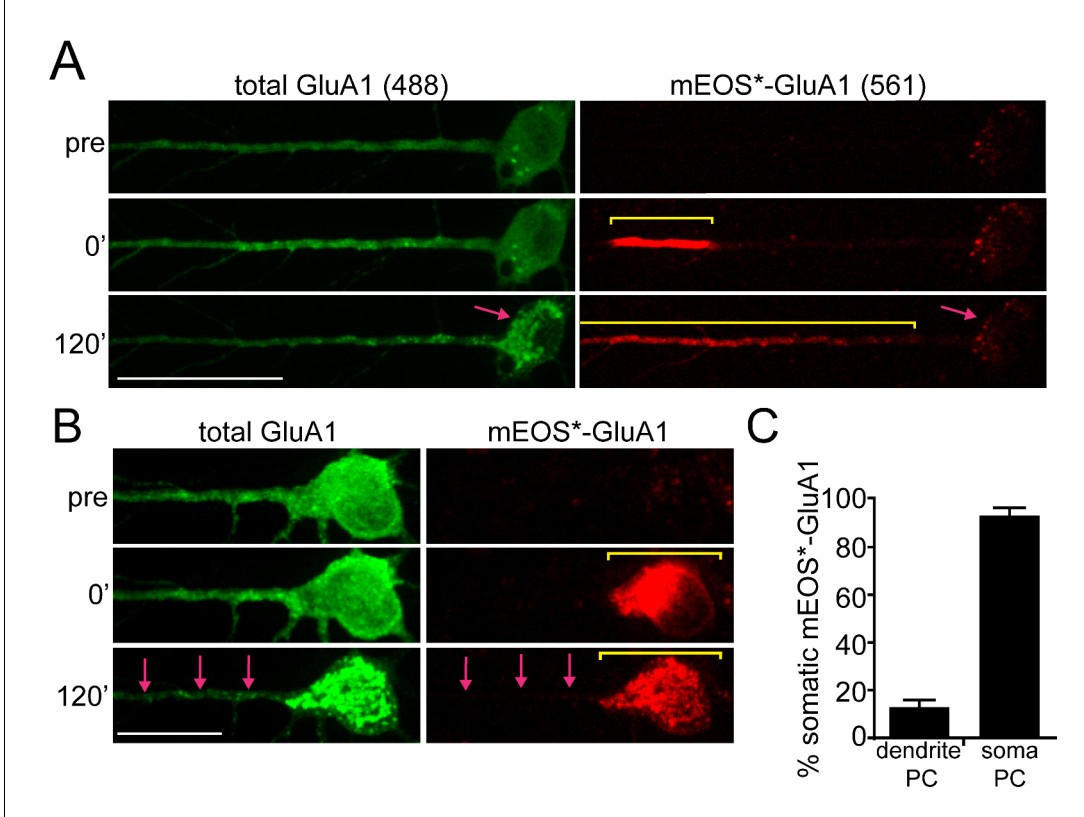

**Figure 2.** Compartmentalization of GluA1 trafficking between the soma and dendrites. (**A**) Localization of dendritically photoconverted mEOS*-GluA1 after ER-release. A cortical neuron expressing 3xF_M-mEOS3.2-GluA1 was imaged at baseline (top panels) and immediately after photoconversion of mEOS3.2 from green to red (middle panels or 0' time point). After photoconversion, DDS was added and the cells were imaged for 120 min (bottom panels). The yellow line highlights the distribution of mEOS*-GluA1 before and after addition of DDS. The pink arrow highlights the cell body, which exhibits robust accumulation of non-photoconverted GluA1 (in the green channel), but does not accumulate mEOS*-GluA1 over the same time period. Scale bar, 50 μm. Also see *Video 1*. (**B**) Cortical neuron imaged before and after somatic photoconversion and following ER release. Photoconversion results in robust generation of mEOS*-GluA1 in the neuronal soma (middle panel). Following addition of DDS, mEOS*-GluA1 redistributes to somatic GA with little entry into dendrites (bottom panels). The yellow bar indicates the distribution of photoconverted cargo before and after ER release. The pink arrows indicate the appearance of dendritic GluA1 puncta (green channel) lacking mEOS*-GluA1 (red channel) following release. Dendrites were computationally straightened using ImageJ; scale bar, 25 μm. Also see *Video 2*. (**C**) Quantification of the percent of mEOS*-GluA1 localized to the neuronal soma 120' after ER release following dendritic or somatic photoconversion. (mean ± SEM, n = 7 neurons for somatic photoconversion; n = 9 neurons for dendritic photoconversion from 3 experiments).

DOI: https://doi.org/10.7554/eLife.27362.007

The following figure supplement is available for figure 2:

**Figure supplement 1.** Photoconversion of 3xF_M-mEOS-GluA1 without ER release.

DOI: https://doi.org/10.7554/eLife.27362.008

## Dendritic recycling endosomes mediate anterograde trafficking in dendrites

We next investigated the fate of secretory cargo following ERGIC exit. There is some evidence that the RE network physically contacts ERGIC membranes in kidney cells, implying a direct ERGIC to RE trafficking route (*Marie et al., 2009*). In neurons, REs are widely distributed throughout dendrites including within a large fraction of dendritic spines (*Figure 5A*, *Figure 5—figure supplement 1*) (*Cooney et al., 2002*; *Kennedy et al., 2010*; *Lasiecka and Winckler, 2011*), making them well suited to participate in local, post-ERGIC trafficking. To test if these organelle networks could be functionally coupled in dendrites, we first asked whether they are in spatial proximity to one another. We imaged live neurons expressing ERGIC53-GFP along with TfR-mCh, which strongly localizes to REs (*Figure 5A*). Using standard confocal microscopy, we

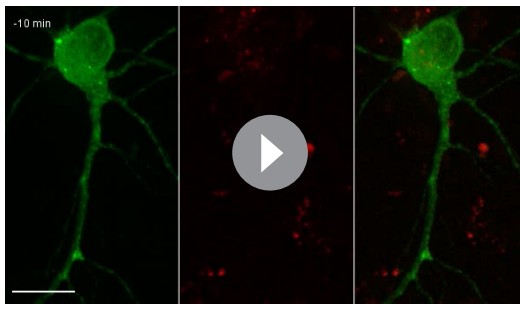

**Video 2.** Spatially restricted trafficking of mEOS-GluA1 photoconverted in the somatic ER. Same as *Video 1* except photoconversion targeted the neuronal soma. Scale bar, 20 μm.
DOI: https://doi.org/10.7554/eLife.27362.011

observed frequent juxtaposition of ERGIC and TfR puncta throughout the dendritic arbor. Despite their high density, this was not simply due to chance as a randomized, but equally dense, set of RE coordinates did not exhibit the same spatial proximity to ERGIC (*Figure 5B–D*).

To gain more detailed structural insight into the relationship between ERGIC and the RE network, we employed super resolution light microscopy using stimulated emission depletion (STED) microscopy (*Hell and Wichmann, 1994*). We labeled neurons expressing TfR-mCh and ERGIC53-GFP with antibodies against GFP and mCh. We observed ERGIC structures, ranging in diameter from ~60 nm (the resolution limit in the far-red channel, *Figure 5—figure supplement 2*) to ~400 nm in diameter, widely distributed throughout dendrites (*Figure 5E*, *Figure 5—figure supplement 3*). ERGIC membranes were

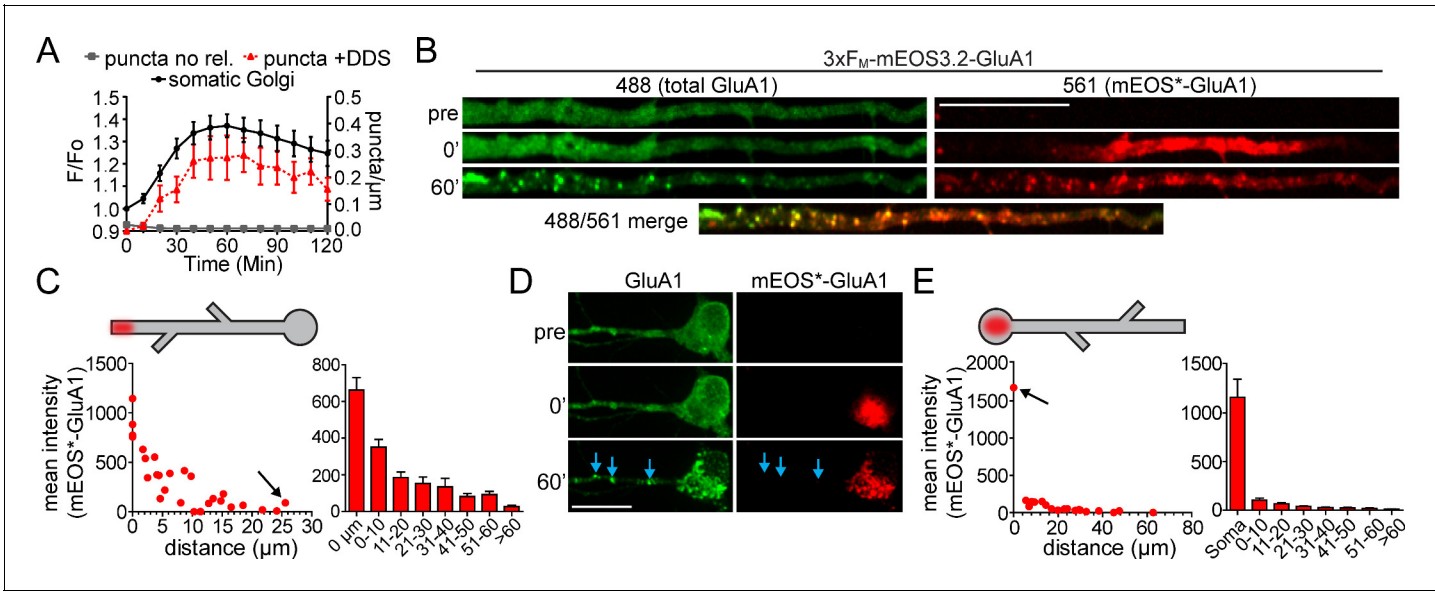

**Figure 3.** Quantification of spatial confinement after GluA1 exits the somatic or dendritic ER. (A) Time-course of the accumulation of GluA1 (total signal in the 488 channel) in the somatic Golgi (mean ±SEM. black line, plotted on left Y-axis, n = 19 neurons, 4 experiments) and the appearance of dendritic GluA1 puncta with (+DDS, red triangles w/dotted line, n = 8 neurons from 3 experiments) or without addition of DDS (puncta no release, grey squares, n = 5 neurons from 2 experiments) plotted on the right Y-axis. (B) Spatially constrained trafficking of photoconverted mEOS-GluA1 to local dendritic organelles following ER release. ER-localized 3xF$_M$-GluA1-mEOS3.2 was photoconverted from green to red in a dendritic segment prior to the addition of DDS. Images show mEOS*-GluA1 signal prior to photoconversion (top), immediately following photoconversion (middle) and 60 min following ER release (bottom). Dendrites computationally straightened in ImageJ. Scale bar, 30 μm. (C) The left panel plots the mean mEOS*-GluA1 fluorescence intensity in individual dendritic puncta as a function of their distance from the boundaries of the photoconversion site, 60 min following ER release. The arrow denotes the somatic Golgi data point. The right panel shows pooled and binned data for 5 neurons, mean ±SEM from 3 experiments. Schematized neuron aligns with X-axis of graph. (D) ER-localized 3xF$_M$-GluA1-mEOS3.2 was photoconverted from green to red in the neuronal soma prior to the addition of DDS. Images show mEOS*-GluA1 signal prior to photoconversion (top), immediately following photoconversion (middle) and 60 min following ER release (bottom). Dendrites computationally straightened in ImageJ. Scale bar, 30 μm. Blue arrows indicate dendritic puncta that appear in the total GluA1 channel, but do not contain detectable mEOS*-GluA1. (E) Quantification of somatically photoconverted GluA1 60 min following ER release. The left graph plots mEOS*-GluA1 signal intensity as a function of distance from the soma (somatic Golgi indicated by the black arrow). The right panel shows pooled and binned data for 5 neurons, mean ± SEM from 3 experiments. Schematized neuron aligns with X-axis of graph.
DOI: https://doi.org/10.7554/eLife.27362.009

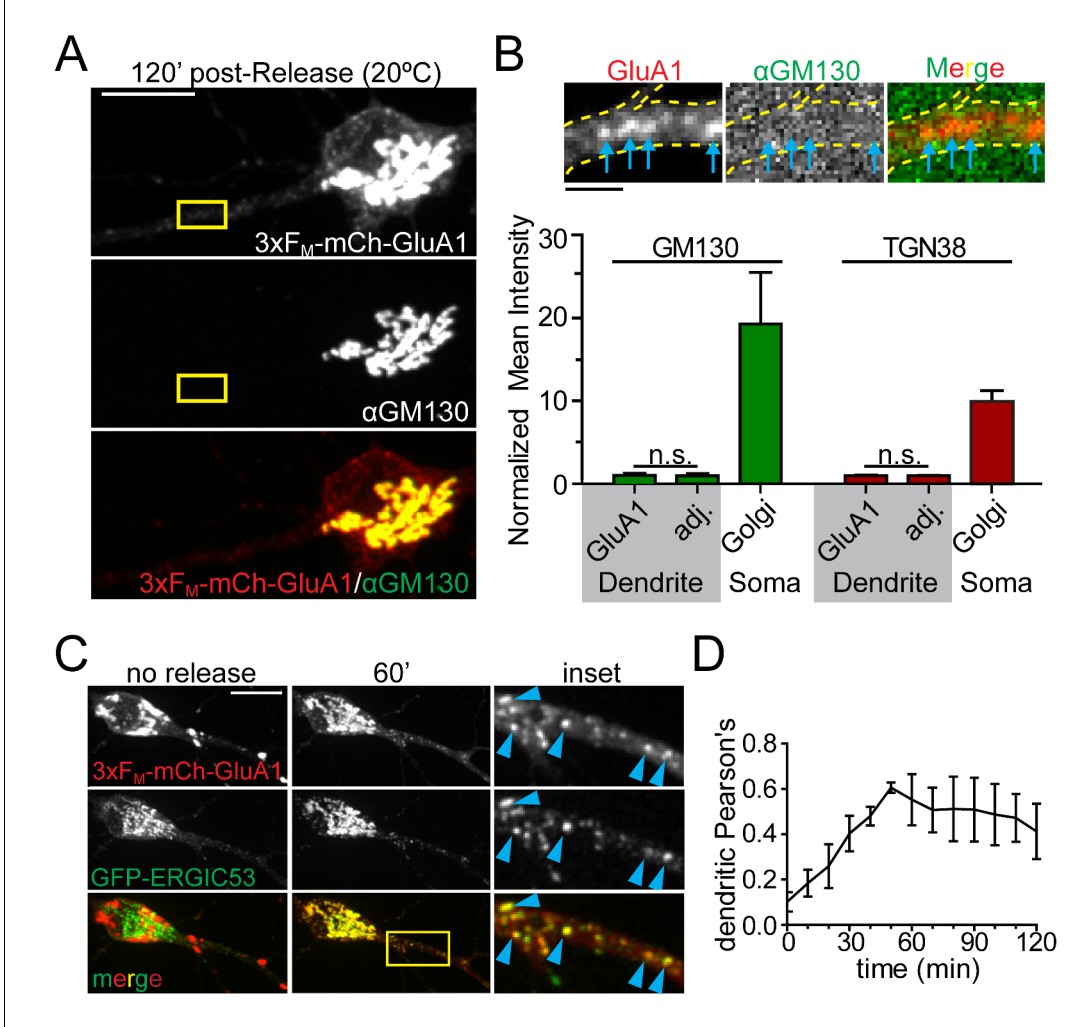

**Figure 4.** GluA1 accumulates in dendritic ER-Golgi intermediate compartments. (A) Dendritic trafficking organelles are negative for canonical GA markers. Shown is GM130 staining (middle panel) of neurons expressing 3xF$_M$-GluA1-mCh (top panel) 120 min after ER-release at 20°C. Scale bar, 10 μm. (B) Images are from the inset in A and show the accumulation of 3xF$_M$-GluA1-mCh in dendritic puncta (blue arrows) that contain no detectable GM130 signal (the brightness of these images has been linearly adjusted to visualize lack of GM130 signal in dendrites). GluA1 puncta (blue arrows) do not stain with GM130. Bottom graph shows quantification of *cis*- (GM130) and *trans*- (TGN38) Golgi markers at GluA1 puncta that form following ER release at 20 °C. The intensities of Golgi-marker staining at GluA1 puncta are compared to immediately adjacent dendritic ROIs negative for GluA1-positive trafficking organelles. Relative intensities of GM130 and TGN38 in the somatic Golgi are also plotted for comparison (mean ±SEM, n = 5 neurons/condition from 2 experiments, n.s. = not significant by unpaired two-tailed Student's t-test.). Scale bar, 2 μm. (C) Colocalization of 3xF$_M$-mCh-GluA1 and GFP-ERGIC53 before and 60 min after addition of DDS. Blue arrowheads denote colocalized dendritic puncta. (D) Colocalization between 3xF$_M$-GluA1-mCh and ERGIC53-GFP within the dendrite was calculated using Pearson's correlation and plotted as a function of time following ER release (mean ± SEM, n = 5 neurons from 2 experiments).

DOI: https://doi.org/10.7554/eLife.27362.012

The following figure supplements are available for figure 4:

**Figure supplement 1.** Dendritic GluA1 puncta are negative for TGN38.

DOI: https://doi.org/10.7554/eLife.27362.013

**Figure supplement 2.** Dendritic localization of ERGIC53.

DOI: https://doi.org/10.7554/eLife.27362.014

**Figure supplement 3.** Colocalization between trafficking GluA1 and an endogenous ERGIC marker.

DOI: https://doi.org/10.7554/eLife.27362.015

**Figure supplement 4.** Post-GA transport from the soma to dendrites.

DOI: https://doi.org/10.7554/eLife.27362.016

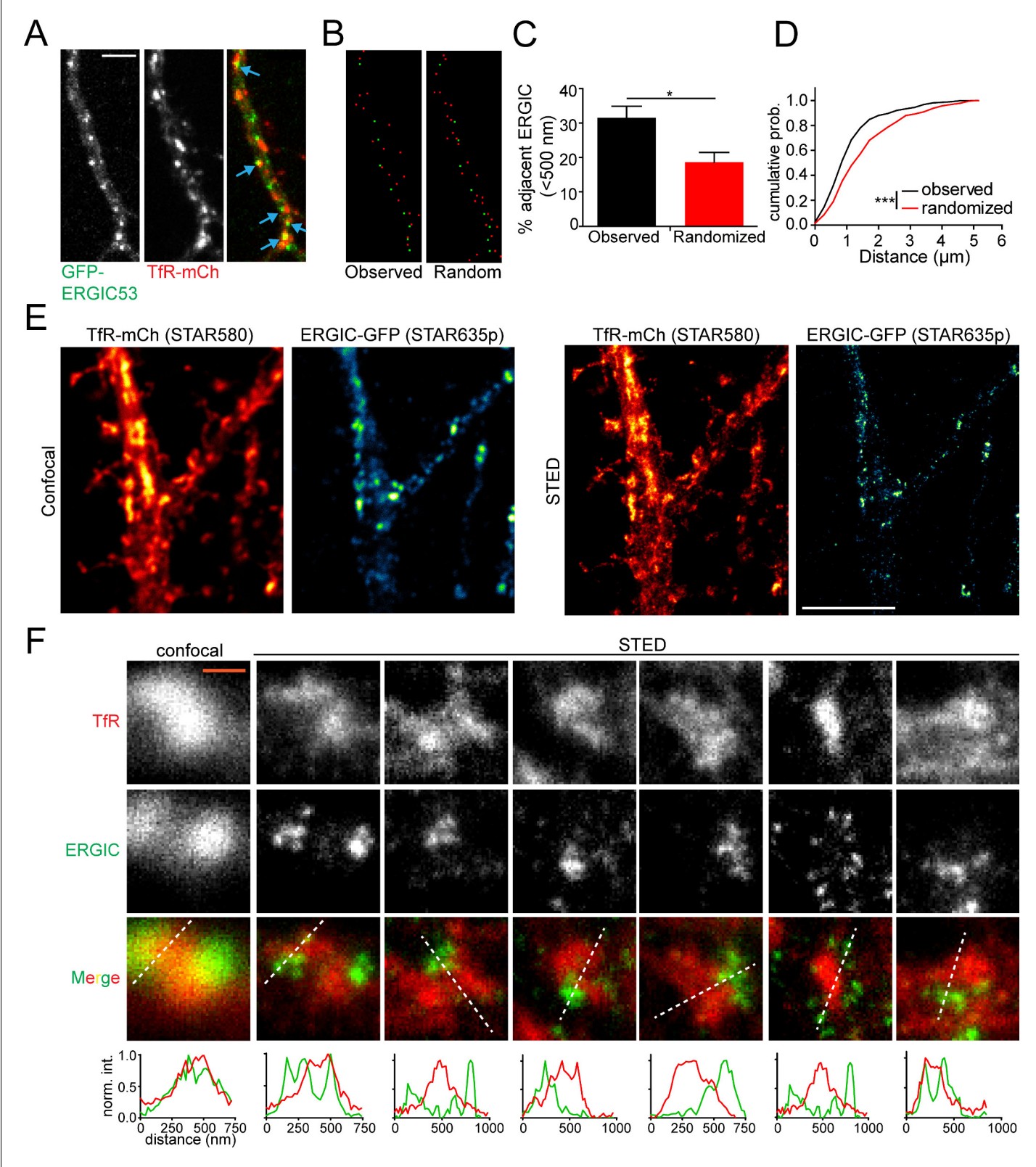

**Figure 5.** REs are located in close proximity to dendritic ERGICs. (**A**) Shown is a section of dendrite from a cultured cortical neuron coexpressing GFP-ERGIC53 and TfR-mCh. Blue arrows indicate adjacent/overlapping puncta. Scale bar, 5 μm. (**B**) The coordinates of the maximum pixel intensity for identified TfR (red) and ERGIC53 (green) puncta are shown. The right panel displays an equally dense but randomized set of TfR coordinates. (**C**) Quantification of the percentage of ERGIC puncta that have at least one TfR punctum within 500 nm relative to a randomized data set with an equal

*Figure 5 continued on next page*

Figure 5 continued

density. (mean ± SEM, n = 8 neurons from 2 experiments *p=0.011 by unpaired two-tailed Student's t-test). (D) The cumulative distribution of the minimum distance between each TfR-positive puncta and the nearest ERGIC puncta (***p<0.0001 Kolmogorov-Smirnov test). (E) Confocal (left pair) and STED (right pair) imaging of cortical neuron expressing TfR-mCh/ERGIC53-GFP stained with α-mCh (2° STAR580) and α-GFP (2° STAR635p). Scale bar, 5 μm. (F) Several examples showing TfR and ERGIC spatial coordination. White lines indicate location for intensity profiles plotted below. Intensity profiles were normalized to the maximum value under each line. The left panels show the confocal images for the first STED example for comparison. Scale bar, 400 nm.

DOI: https://doi.org/10.7554/eLife.27362.017

The following figure supplements are available for figure 5:

**Figure supplement 1.** Spine and dendritic localization of REs.

DOI: https://doi.org/10.7554/eLife.27362.018

**Figure supplement 2.** Characterization of STED FWHM.

DOI: https://doi.org/10.7554/eLife.27362.019

**Figure supplement 3.** Segmentation analysis of organelle size from STED images.

DOI: https://doi.org/10.7554/eLife.27362.020

often observed immediately adjacent to TfR-positive structures (*Figure 5F*), consistent with the possibility of functional coupling.

Given the spatial proximity of ERGIC and REs, we next asked whether $3xF_M$-GluA1 enters the dendritic RE network on its journey to the PM. We imaged live neurons expressing $3xF_M$-mCh-GluA1 along with TfR-GFP before and after ER release. We observed robust colocalization of TfR and GluA1 120 min following addition of DDS (*Figure 6A*). To more rigorously establish trafficking of recently released cargo through REs, we employed a previously described photobleaching assay that allows us to visualize mobile intracellular vesicles with high signal to noise (*Esteves da Silva et al., 2015*). We co-expressed $3xF_M$-mCh-GluA1 along with either GFP-Rab11 or TfR-GFP to label REs. We photobleached both the mCh (GluA1) and GFP (RE marker) signals within ~50 μm segments of dendrite (*Figure 6B*). New vesicles entering the bleached field from neighboring unbleached regions can be easily visualized in both channels, allowing us to unambiguously assign co-trafficking of recently released mCh-GluA1 and RE marker proteins (*Figure 6C*, *Video 3*). Before ER-release, numerous mobile REs could be observed, but these were almost never (1/77 for Rab11-labeled REs, 0/60 for TfR-labeled REs) positive for $3xF_M$-mCh-GluA1 (*Figure 6D,E*). In contrast, one hour following ER-release, 32.2 ± 5.5% of GluA1-containing vesicles co-traffic with TfR-GFP and 37.1 ± 5.5% co-traffic with GFP-Rab11. After 150 min, these numbers increased to 52.9 ± 6.7% for TfR-GFP and 59.9 ± 4.3% for GFP-Rab11 (*Figure 6D,E*, *Video 4*; mean ± SEM, n = 6–8 neurons/timepoint/marker from 3 experiments).

Even though minimal GluA1 had reached the cell surface at time points where we observe robust RE accumulation (*Figure 1D*), we wanted to rule out the possibility that GluA1 had been delivered to the cell surface and endocytosed into the RE network (*Ehlers, 2000*). To do so, we utilized the extracellular thrombin cleavage site in $3xF_M$-mCh-GluA1 located between mCh and GluA1 and performed our co-trafficking assay in the presence of extracellular thrombin (*Figure 1A*, *Figure 1—figure supplement 2*, *Figure 1—figure supplement 3*). Because thrombin quickly removes the fluorescent tag from surface GluA1 (*Figure 1—figure supplement 3*), we would expect to observe a reduction in mCh-GluA1/GFP-Rab11 cotrafficking if prior surface trafficking is required for entry into the RE network. However, thrombin had no effect on the extent of Rab11/GluA1 co-trafficking following ER release supporting a prominent role for REs in forward trafficking in dendrites (*Figure 6E*).

It remains formally possible that GluA1 could be inserted into the PM and then quickly endocytosed into the RE network before thrombin acts. While this possibility is unlikely given the slow basal rate of AMPA receptor internalization (*Ehlers, 2000*; *Passafaro et al., 2001*), we confirmed anterograde RE trafficking using an entirely different strategy. We engineered a soluble anterograde trafficking marker consisting of mCh fused to four $F_M$ domains ($4xF_M$-mCh). It is impossible for this cargo to recycle since it is released into the media following exocytosis (*Figure 6F*, *Figure 6—figure supplement 1*). We observed nearly identical co-trafficking of $4xF_M$-mCh and GFP-Rab11 as we did with $3xF_M$-mCh-GluA1 and GFP-Rab11 (48.9 ± 3% of Rab11 vesicles contained $4xF_M$-mCh 2.5 hr post-release; mean ± SEM; n = 8 neurons from 2 experiments), indicating that RE-mediated forward

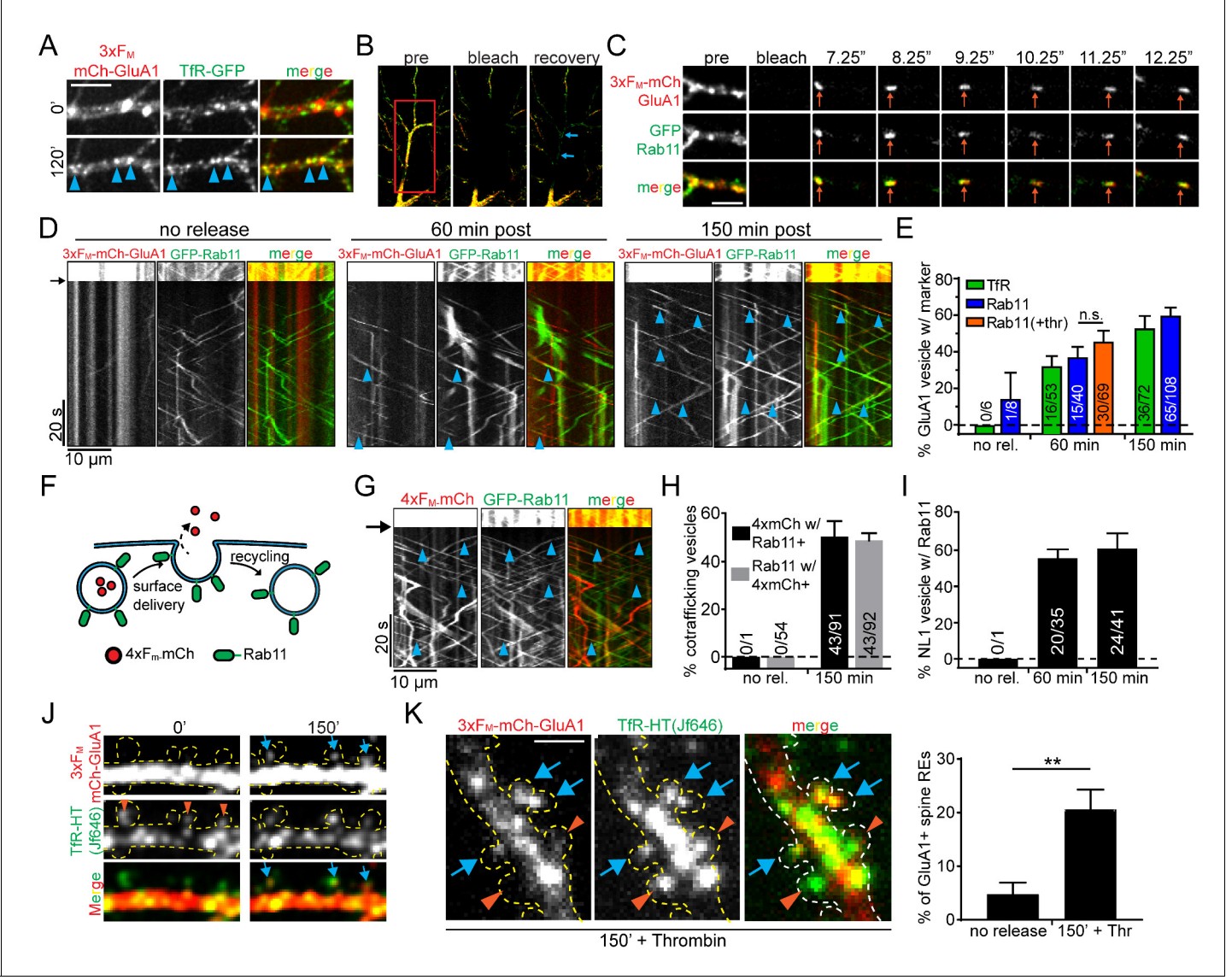

**Figure 6.** Recycling endosomes mediate anterograde trafficking in dendrites. (**A**) Live-cell imaging of a cortical neuron expressing 3xF$_M$-GluA1-mCh and TfR-GFP before ER-release (top panels) and 120 min after ER-release (bottom panels). Blue arrowheads indicate locations where GluA1 has redistributed to TfR-GFP positive endosomes. Scale bar, 10 μm. (**B**) Experimental paradigm to visualize individual vesicles trafficking in neurons expressing 3xF$_M$-GluA1-mCh and a green RE marker (either GFP-Rab11 or TfR-GFP). The middle panel shows photobleaching of a dendritic segment (indicated by red square). The right panel is a frame taken 20 s after photobleaching showing endosomes entering the bleached area (blue arrows) Also see *Videos 3* and *4*. (**C**) Representative example of a mobile vesicle (orange arrows) associated with both Rab11 and 3xF$_M$-GluA1. Individual frames are taken from 4 Hz dual-color imaging of a dendritic segment following photobleaching. Scale bar, 5 μm. (**D**) Kymographs of dendritic segments following photobleaching showing the movement of Rab11 and GluA1 vesicles at various times following 3xF$_M$-mCh-GluA1 release from the ER. The amount of time elapsed between addition of DDS and imaging is indicated above the kymographs. The black arrow indicates the time of photobleaching. Blue arrowheads denote double-positive mobile vesicles. (**E**) Quantification of the percentage of GluA1 vesicles that also contain the indicated RE marker. The orange bar (+Thr) indicates the inclusion of 1 U/ml thrombin along with DDS. (mean ± SEM, n = 6–8 neurons/timepoint/marker from 3 experiments per marker; 2 independent experiments for thrombin n.s. p=0.31 by unpaired two-tailed Student's t-test). Numbers on each bar indicate the raw number of double-positive vesicles/total GluA1 vesicles. (**F**) Schematic of soluble anterograde trafficking marker (4xF$_M$-mCh). During vesicle exocytosis, the soluble marker is released from the cell and therefore cannot be recycled. (**G**) Kymographs showing cotrafficking between anterograde soluble marker (4xF$_M$-mCh) and GFP-Rab11 within a segment of dendrite 150 min after DDS addition. Blue arrowheads highlight individual cotrafficking vesicles. (**H**) Quantification of the percentage of vesicles cotrafficking Rab11 and 4xF$_M$-mCh before and 150 min after ER-release (mean ± SEM, no rel. n = 5 neurons; 150 minutes n = 8 neurons from 2 experiments). (**I**) Quantification of cotrafficking between NL1 and Rab11. Numbers on bars indicate raw numbers of vesicles positive for both markers divided by the total number positive for NL1 (mean ± SEM, n = 5–6 neurons/condition from 2 experiments). (**J**) GluA1 accumulates in a subset of spine endosomes following ER release. Shown is a stretch of dendrite from a neuron expressing 3xF$_M$-mCh-GluA1 along with TfR-HaloTag labeled with JF646 before and 150 min following addition of DDS. Orange arrowheads denote spines

*Figure 6 continued on next page*

*Figure 6 continued*

containing REs. Blue arrows indicate accumulation of GluA1 in spine-resident REs. Dotted yellow outline drawn based on GFP cell fill (not shown). (K) Images are of a neuron coexpressing 3xF$_M$-mCh-GluA1 and TfR-HaloTag (JF646) fixed 150 min after addition of DDS in the presence of thrombin (1 U/ml) to prevent visualization of recycled proteins. As in F, blue arrows highlight spine REs that contain trafficking GluA1. Orange arrowheads indicate GluA1-negative spine REs. The right panel is the quantification of the percentage of spines with REs that also contain GluA1 before and 150' after release in the presence of thrombin. (mean ± SEM, n = 5 neurons/condition from 2 experiments, **p=0.006 unpaired two-tailed Student's t-test.) Scale bar, 2 µm.

DOI: https://doi.org/10.7554/eLife.27362.021

The following figure supplements are available for figure 6:

**Figure supplement 1.** Verification of soluble anterograde trafficking marker.

DOI: https://doi.org/10.7554/eLife.27362.022

**Figure supplement 2.** Fraction of GluA1 insertion events in the soma and dendrites.

DOI: https://doi.org/10.7554/eLife.27362.023

**Figure supplement 3.** Surface GluA1 is increased at spines with REs compared to neighboring RE-negative spines.

DOI: https://doi.org/10.7554/eLife.27362.024

**Figure supplement 4.** Relationship between the spine apparatus and REs.

DOI: https://doi.org/10.7554/eLife.27362.025

trafficking in dendrites applies to soluble as well as integral membrane secretory cargoes (*Figure 6G,H*). We further confirmed the generality of this trafficking route using a different synaptic protein NL1, which also entered the RE network soon after ER-release (*Figure 6I*). The comparatively rapid entry of NL1 into the RE network is consistent with its accelerated surface delivery (*Figure 1E*) relative to GluA1 (*Figure 1D*). Finally, we determined the location of newly trafficked GluA1 insertion into the neuronal PM by visualizing discrete insertion events using the pH-sensitive fluorescent protein superecliptic pHluorin (SEP) fused to GluA1. Because SEP is quenched in the acidic RE lumen, discrete exocytosis events can be observed as the RE fuses with the PM and quickly neutralizes (*Yudowski et al., 2007*). We included extracellular thrombin in this experiment to remove the SEP tag from the surface pool of receptors to limit the possibility of visualizing re-insertion of recycled receptors. We observed that 67 ± 8% of fusion events occurred in the dendritic arbor vs. 33 ± 8% in the soma (*Figure 6—figure supplement 2*).

## GluA1 accumulates in a fraction of spine endosomes following ER release

Consistent with previous reports, we observed REs throughout dendrites, including within 48% of dendritic spines, the major postsynaptic compartments of excitatory synapses (*Figure 5—figure supplement 1*) (*Cooney et al., 2002*; *Kennedy et al., 2010*). In contrast, ERGIC membranes only rarely localize to spines (*Figure 4—figure supplement 2*, *Figure 5—figure supplement 1*). During live-cell imaging of 3xF$_M$-mCh-GluA1 and TfR-HaloTag (TfR-HT) labelled with HaloTag ligand JF646 (HTL-646) (*Grimm et al., 2015*), we noted the accumulation of 3xF$_M$-mCh-GluA1 in spine REs that initially lacked GluA1 signal (*Figure 6J*). We quantified the fraction of RE-positive spines that gained GluA1 signal 150 min following ER release in extracellular solution containing thrombin to eliminate recycling receptors from our analysis. We found that 3xF$_M$-mCh-GluA1 was detectable within 20.6 ± 3.7% of RE-positive spines vs only 4.8 ± 2.1% in control (no ER release) conditions (*Figure 6K*; mean ± SEM, n = 5 neurons/condition from 2

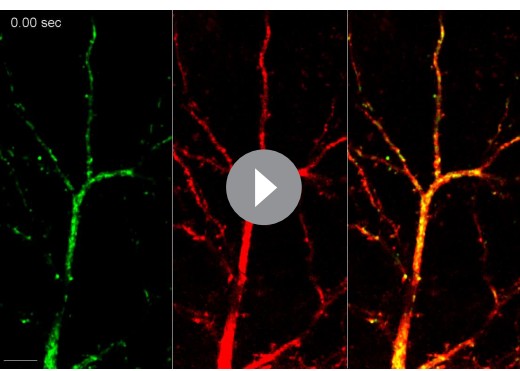

**3.** Photobleaching assay for tracking secretory cargo. Photobleaching assay in a neuron expressing GFP-Rab11 and 3xF$_M$-mCh-GluA1 150' following ER release. Left panel shows GFP-Rab11, middle shows 3xF$_M$-mCh-GluA1 and right shows the merge. Rapid (4 Hz), single plane imaging was carried out before and after photobleaching. Scale bar, 20 µm.

DOI: https://doi.org/10.7554/eLife.27362.026

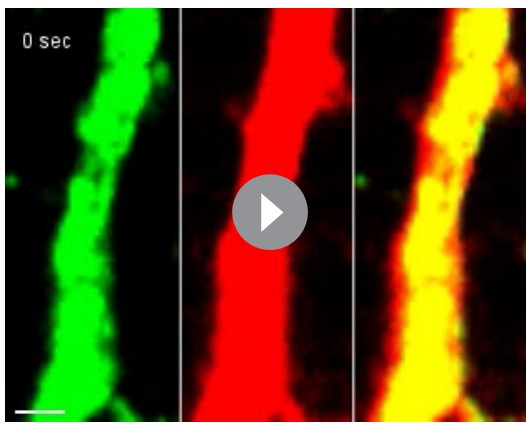

**Video 4.** Forward trafficking GluA1 accumulates in REs. Zoomed in section of dendrite from the photobleaching assay shown in *Figure 6* and *Video 3*. A high proportion of mobile vesicles harbored both GFP-Rab11 (green) and 3xF$_M$-mCh-GluA1 (red) 150' following ER release. Scale bar, 2 µm.
DOI: https://doi.org/10.7554/eLife.27362.027

experiments, p=0.006 unpaired two-tailed Student's t-test.). Accordingly, when we surface labeled mCh-GluA1 receptors 120 min following ER release, we observed that spines harboring REs accumulated more surface GluA1 than neighboring spines lacking REs (*Figure 6—figure supplement 3*) consistent with a role for REs in local delivery of secretory cargo. Finally, we tested whether there was any relationship between excitatory synapses with a spine apparatus, a membrane-stacked organelle hypothesized to play a GA-like function within spines, and REs. We expressed synaptopodin, a molecular marker for the spine apparatus, fused to GFP (GFP-SP), which displayed very punctate localization within or at the base of ~25% of spines consistent with previous reports (*Figure 6—figure supplement 4A*) (*Vlachos et al., 2009*; *Korkotian et al., 2014*). Intriguingly, nearly every (97 ± 3%; mean ± SEM, 554 RE-containing spines from 4 neurons) dendritic spine that contained GFP-SP also contained an RE marked by TfR-mCh (although not all RE-containing spines contained GFP-SP). In many cases the two labels were resolvable by confocal microscopy suggesting that they localize to distinct organelles (*Figure 6—figure supplement 4B*). Together, these data support a model where REs not only relay cargo to the dendritic surface but can also direct newly synthesized cargo to select synapses, perhaps directly receiving cargo from the ER-derived spine apparatus.

## Disruption of RE function impairs GluA1 ER to PM trafficking

To further support a role for REs in dendritic anterograde secretory trafficking, we disrupted RE function using a dominant-negative version of the RE-associated small GTPase, Rab11 (S25N mutation). Rab11(S25N) significantly impaired surface delivery of 3xF$_M$-GluA1 compared to Rab11(WT) or mCh alone at the earliest time points we can detect appreciable surface GluA1 accumulation (2 and 4 hr after DDS addition) (*Figure 7A,B*). While Rab11(S25N) robustly impaired surface delivery, it did not completely block it, suggesting that additional trafficking routes could mediate surface expression. To test whether the more canonical post-GA trafficking pathway (which depends on Rab8) is also utilized, we performed the same trafficking experiment using dominant negative Rab8 (T22N mutation). We observed that Rab8(T22N) did not significantly affect GluA1 surface delivery at an early timepoint (2 hr) following ER release, but significantly impaired surface delivery at a later (4 hr) timepoint (*Figure 7C*). This result suggests that Rab11-mediated trafficking plays a predominant role in early surface delivery while both Rab8 and Rab11 pathways participate in later phases of surface delivery. Finally, we tested dominant negative Rab5 (S34N mutation), which is critical for early endosome function, and observed no effect on surface delivery (*Figure 7D*). Combined, these data strongly support a major functional role for the dendritic RE network in forward secretory trafficking.

## Golgi-independent transfer of GluA1 to the PM through ERGIC and REs

Anterograde trafficking through REs has been established as a major post-GA trafficking route (*Ang et al., 2004*; *Farr et al., 2009*). However, since the majority of dendrites lack canonical GA, we next tested whether secretory cargoes in dendrites can bypass the GA on their route to REs and the PM. We performed our trafficking assay in the presence of 5 µg/mL BFA, which potently disrupts Golgi membranes (*Fujiwara et al., 1988*). To verify the efficacy of our BFA treatment, we imaged live neurons expressing galactosyl transferase fused to mEmerald (GalT-mEm). BFA caused a rapid (<20 min) and complete dispersal of the somatic Golgi (*Figure 8A*). In contrast, ERGIC membranes and REs remained intact following BFA treatment (*Figure 8B–D*). In addition to disrupting GA

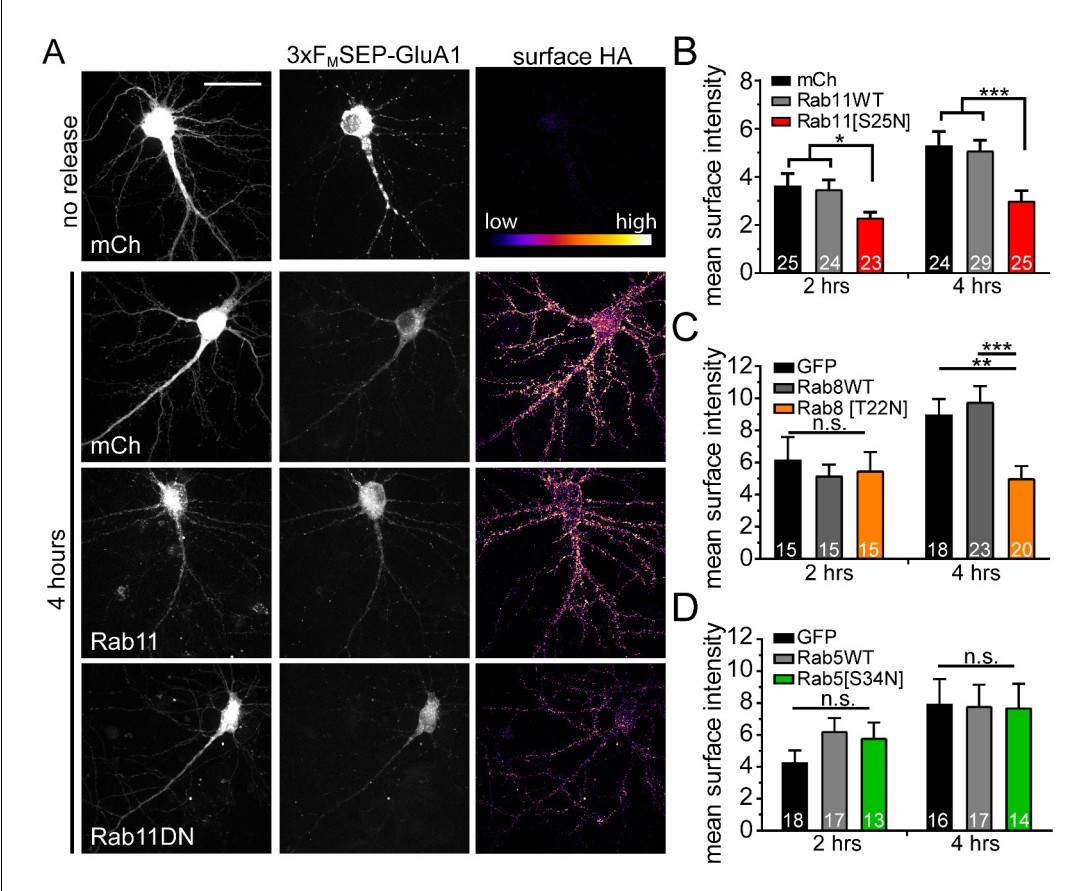

**Figure 7.** Disrupting recycling endosomes, but not early endosome function, inhibits surface GluA1 delivery. (**A**) Surface delivery of 3xF$_M$-SEP-GluA1 two or four hours after addition of DDS in cells co-expressing mCh, mCh-Rab11WT or mCh-Rab11DN. Scale bar, 50 μm. (**B**) Quantification of mean surface GluA1 intensity (assayed by surface staining for HA-epitope as in A) for Rab11-variant expressing neurons (mean ± SEM, n = number of neurons displayed on bars from 4 independent experiments, *p=0.0239 (mCh) and 0.0129 (Rab11WT) at 2 hr; *** p 0.0021 (mCh) 0.0012 (Rab11WT) at 4 hr by unpaired two-tailed Student's t-test; error bars represent SEM). All values were normalized to the no release condition. (**C**) Quantification of mean surface GluA1 intensity (assayed by surface staining for HA-epitope as in A) for Rab8-variant expressing neurons (mean ± SEM, n = number of neurons displayed on bars from 3 independent experiments, **p=0.0026, ***p=0.0006, n.s. = not significant by unpaired two-tailed Student's t-test; error bars represent SEM). All values were normalized to the no-release condition. (**D**) Surface delivery of GluA1 in neurons co-expressing GFP, GFP-Rab5WT or GFP-Rab5DN (mean ± SEM, n values displayed on bars from 3 independent experiments, n.s. by unpaired two-tailed Student's t-test). All values were normalized to the no-release condition.

DOI: https://doi.org/10.7554/eLife.27362.028

function, BFA is known to block ER-export (*Misumi et al., 1986*; *Lippincott-Schwartz et al., 1989*), which precluded us from simply adding BFA prior to ER release to test for a GA-independent trafficking network in neurons. To circumvent this issue, we took advantage of the temperature sensitivity (*Figure 8—figure supplements 1* and *2*), but BFA insensitivity (*Figure 8C,D*), of dendritic ERGICs. We first released 3xF$_M$-mCh-GluA1 from the ER for 120 min at 20 ˚C to allow cargo accumulation in somatic GA and dendritic ERGICs (*Figure 8E*). While we observed robust accumulation of 3xF$_M$-mCh-GluA1 in ERGIC at 20 ˚C (*Figure 8—figure supplement 1*), it did not progress to REs or to the PM, confirming the efficacy of the temperature block (*Figure 8—figure supplements 1* and *2*). We then added BFA following ERGIC/GA accumulation at 20 ˚C and incubated cells for an additional 20 min to disrupt GA before warming the cells to 37 ˚C (*Figure 8E*). In control experiments using HeLa cells, this protocol completely blocked delivery of secretory cargo to the PM, in agreement with previous work (*Miller et al., 1992*) (*Figure 8—figure supplement 3*). In contrast, when we conducted the same experiment in cortical neurons, we observed robust co-trafficking of 3xF$_M$-mCh-GluA1 and the RE marker TfR-GFP, followed by the delivery of a significant fraction of GluA1

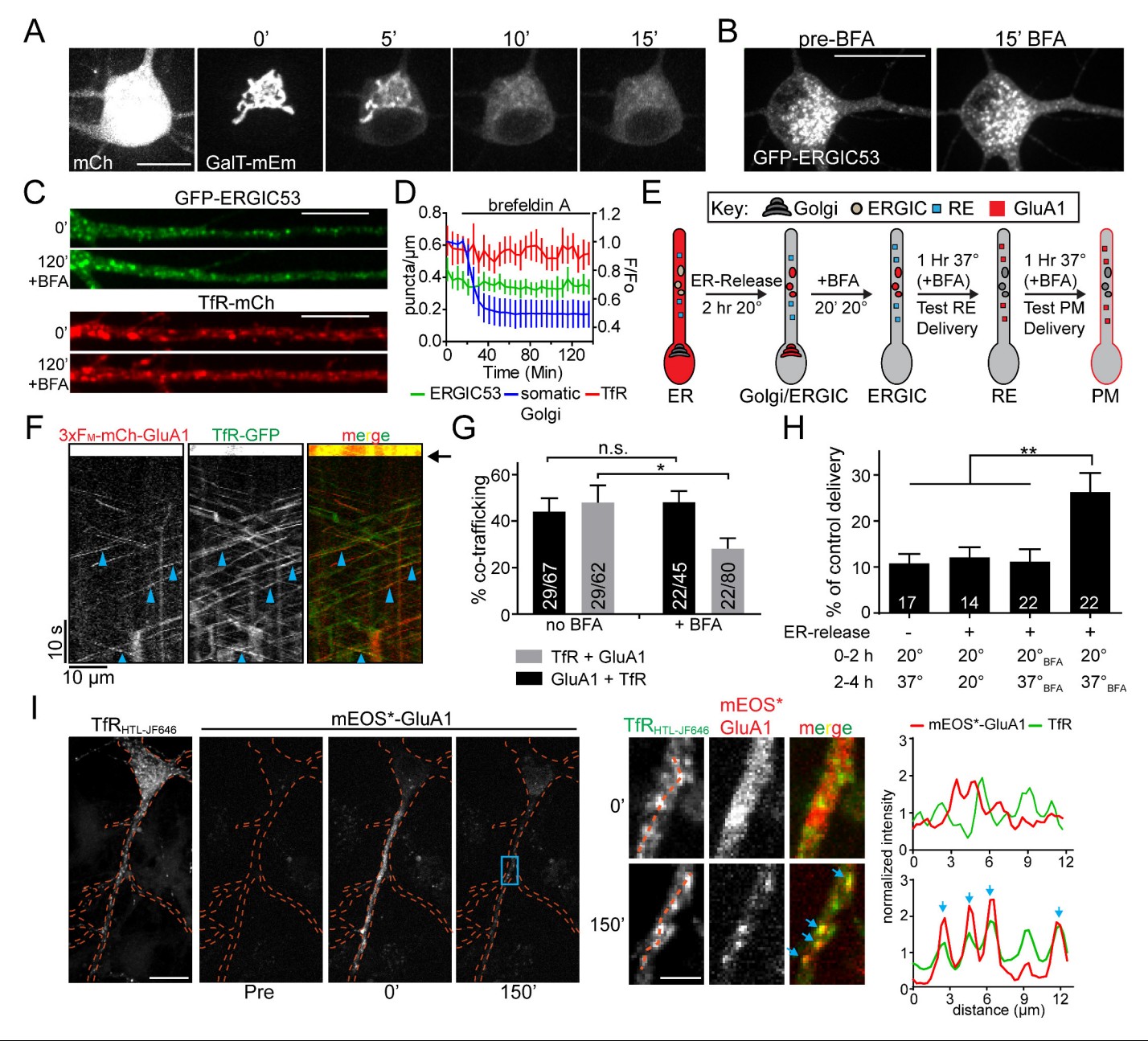

**Figure 8.** Golgi-independent trafficking of GluA1 in dendrites. (**A**) The Golgi apparatus (labelled with galactosyl-transferase fused to mEmerald) rapidly disperses upon BFA (5 μg/ml) application. Scale bar, 15 μm. (**B**) Distribution of somatic GFP-ERGIC53 before and 15 min after application of BFA. Scale bar, 25 μm. (**C**) Dendritic segments from neurons expressing GFP-ERGIC53 (top pair) or TfR-mCh (bottom pair) before and 120' after the application of BFA. Scale bar, 15 μm. Dendrites computationally straightened in ImageJ. (**D**) Quantification of the number of TfR (red) and ERGIC53 (green) puncta after treatment with BFA (mean ± SEM, n = 7 neurons from 2 independent experiments) plotted on the left axis. The dispersal of fluorescent signal from the somatic Golgi is plotted on the right axis normalized to the pre-BFA value (blue, mean ± SEM, n = 5 neurons from 2 independent experiments). (**E**) Schematic of the experimental design to investigate Golgi-independent trafficking. Organelle distribution of GluA1 is highlighted in red. Before ER-release, GluA1 is distributed throughout the ER (ER). 2 hr after addition of DDS at 20°C GluA1 accumulates in the dendritic ERGIC and somatic Golgi compartments (ERGIC/Golgi). Addition of BFA disrupts the somatic Golgi, but leaves dendritic ERGIC intact (ERGIC). Cells are returned to 37°C and after a 1 hr incubation, RE-localization is assessed (panels F,G). 2 hr after returning to 37°C, cells are surface labeled to assess membrane delivery (panel H). (**F**) Shown is a kymograph from a dendritic segment showing cotrafficking between mCh-GluA1 and TfR-GFP in the presence of BFA, 60 min following return to 37 °C. (see 'RE' panel in diagram from 8E). Black arrow denotes timing of the bleach. Blue arrowheads indicate cotrafficking vesicles. (**G**) Quantification of cotrafficking between GluA1 vesicles and TfR signal (black bars) and TfR vesicles and GluA1 signal (grey bars) after application of BFA (mean ± SEM, n = 5–6 neurons/condition from 3 experiments, *p=0.041 and n.s. p=0.59 based on unpaired two-tailed Student's t-test). Numbers

*Figure 8 continued on next page*

Figure 8 continued

on each bar indicate the total number of vesicles. (**H**) Surface delivery of GluA1 occurs when the GA is disrupted. Surface GluA1 was measured in cells incubated at 20 ˚C, then shifted to 37 ˚C without DDS; cells treated with DDS but maintained at 20 ˚C; cells treated with BFA prior to DDS addition; or cells treated with DDS at 20 ˚C to allow ER release, followed by BFA treatment and 37 ˚C incubation. Values are reported as a percentage of maximal delivery that occurs in neurons treated with DDS, but not treated with BFA; control condition n = 25 neurons. The number of neurons measured for each experimental condition are displayed on the bar graph from 2 independent experiments (values are reported as mean ± SEM; ** p 0.0077, 0.0043 and 0.007 respectively (left to right) by unpaired two-tailed Student's t-test.). (**I**) Cortical neuron coexpressing TfR-HaloTag(JF646) and 3xF$_M$-mEOS3.2-GluA1 before (pre) and immediately after dendritic photoconversion (0') as well as 150 min after addition of DDS. Expanded images (from blue rectangle) showing dendritic mEOS*-GluA1 and TfR-HT localization before and 150 min following ER release are shown to the right. The intensity plots were generated using the dotted orange lines. Blue arrows indicate the overlapping intensity peaks present in both the mEOS*-GluA1 channel and in the TfR channel. Scale bar, 20 μm; Inset scale bar, 5 μm.

DOI: https://doi.org/10.7554/eLife.27362.029

The following figure supplements are available for figure 8:

**Figure supplement 1.** Temperature sensitivity of GluA1 trafficking through ERGIC.

DOI: https://doi.org/10.7554/eLife.27362.030

**Figure supplement 2.** Temperature sensitivity of GluA1 entry into recycling endosomes.

DOI: https://doi.org/10.7554/eLife.27362.031

**Figure supplement 3.** BFA sensitivity of surface trafficking in HeLa cells.

DOI: https://doi.org/10.7554/eLife.27362.032

**Figure supplement 4.** BFA sensitivity of VSV-G and NL1 surface trafficking in cortical neurons.

DOI: https://doi.org/10.7554/eLife.27362.033

to the plasma membrane (**Figure 8G,H**). Thus, ERGIC-localized cargo can access the plasma membrane through the RE network in the absence of GA function. Intriguingly, BFA had no effect on the percentage of dendritic 3xF$_M$-mCh-GluA1 co-labeled with TfR 60 min following the temperature shift to 37 ˚C (**Figure 8G**). Not surprisingly, we did observe a significant reduction in the fraction of TfR vesicles positive for GluA1, indicating that some fraction of RE-localized cargo is derived from BFA-sensitive GA (**Figure 8G**). Accordingly, the level of GluA1 surface delivery was reduced in BFA-treated cells (25.3 ± 3.8% of control cells without BFA), but still 2.4-fold over background levels (**Figure 8H**). Together, these data indicate that in neurons, a substantial fraction of GluA1 enters the RE network and is trafficked to the cell surface in a GA-independent manner.

To determine whether a GA-independent trafficking pathway is generally utilized by all membrane proteins, we tested whether NL1 or a generic trafficking cargo, vesicular stomatitis viral glycoprotein (VSV-G), could also reach the cell surface in the presence of BFA. In contrast to GluA1, we observed that both NL1 and VSV-G surface expression was fully blocked by BFA indicating that not all cargoes follow the same trafficking route in neurons (**Figure 8—figure supplement 4**).

To further support a local, GA bypass model for GluA1 in neuronal dendrites, we employed the same local mEOS photoconversion strategy we used to demonstrate ER to ERGIC trafficking (**Figure 2**). Neurons were transfected with 3xF$_M$-mEOS-GluA1 and TfR-HT and labelled with HTL-JF646. The mEOS signal was converted from green to red in 80–100 μm segments of dendrite, followed by addition of DDS to release GluA1 from the ER (**Figure 8I**). Live imaging over the next 150 min revealed colocalization of mEOS*-GluA1 and TfR-HT(JF646) in the same dendritic branches that were initially photoconverted (**Figure 8I**). Importantly, we observed little accumulation of mEOS*-GluA1 in the somatic GA over the time course of the experiment. Thus, dendritic secretory cargo is spatially constrained as it progresses from ER to ERGIC to REs.

By necessity, our experiments rely on expressed proteins with tags to monitor their progression through the secretory pathway. To assess the fraction of endogenous proteins that utilize a GA-bypass pathway in neurons, we took a biochemical approach. A recent report demonstrated that a significant fraction of select neuronal surface proteins, including AMPA, NMDA and GABA receptors display immature (high mannose) glycosylation profiles, consistent with a GA-independent trafficking route (**Hanus et al., 2016**). High mannose glycans generated in the ER are normally processed in the GA to become resistant to cleavage by the glycolytic enzyme endoglycosidase H (endoH). Thus, to estimate the fraction of protein that circumvents the GA, we purified surface proteins by surface biotinylation and subjected them to either endoH or peptide-N-glycosidase (PNGase, a glycolytic enzyme which removes both mature and immature N-linked carbohydrates) (**Figure 9A,B**). We then

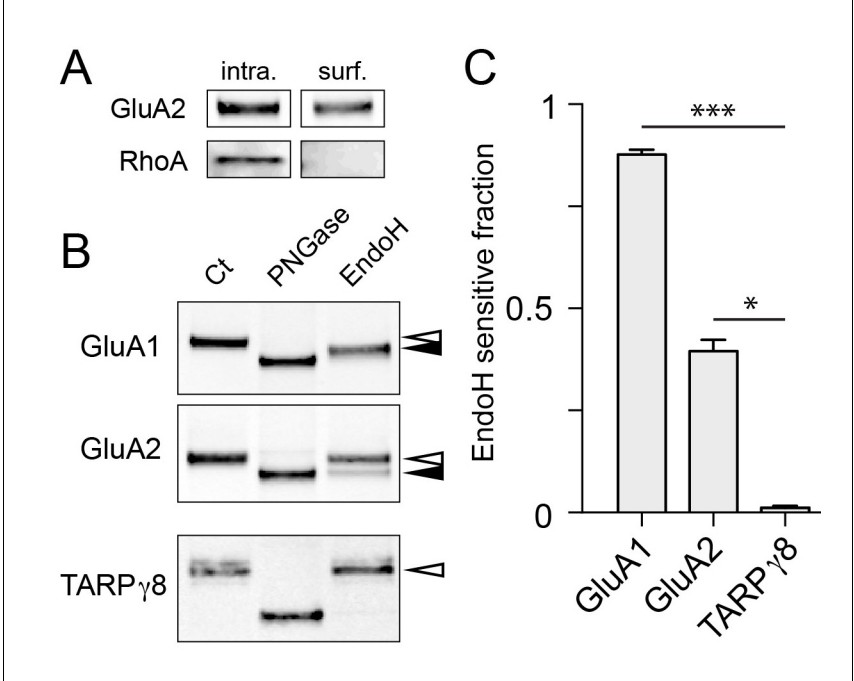

**Figure 9.** Surface AMPA receptor subunits display immature glycosylation. (**A**) Control experiment for surface biotinylation. Shown are immunoblots against a surface AMPA receptor (GluA2) and a cytosolic protein (RhoA) following surface biotinylation and purification over streptavidin. Note the exclusion of RhoA from the surface fraction confirming the integrity of the cells during surface biotinylation. (**B**) Surface AMPA receptors are sensitive to endoH. Shown are immunoblots against surface GluA1 (top), GluA2 (middle), and TARPγ8 (bottom). Purified biotinylated surface fractions were either untreated (first lane) or treated with either PNGase (middle lane) or endoH (third lane). Note the mobility shift of endoH treated GluA1 and GluA2 but not TARPγ8. The black arrowheads denote the endoH-shifted species while the white arrowheads mark the mobility of the untreated native surface protein. (**C**) The fraction of endoH sensitive GluA1, GluA2 and TARPγ8 are plotted. Values are reported as mean ±SEM from 4, 5 and 4 independent experiments for GluA1, GluA2 and TARPγ8 respectively. *, p<0.05; ***, p<0.01, ANOVA and Dunn's multicomparison test.
DOI: https://doi.org/10.7554/eLife.27362.034

compared the SDS/PAGE mobility of enzyme-treated surface GluA1 and GluA2 to an untreated control sample to estimate the fraction of immature surface GluA1 and GluA2. Consistent with previous work, we observed that the majority (88 ± 1%; mean ± SEM 4 independent experiments) of surface GluA1 and a significant fraction of GluA2 (40 ± 3%, mean ± SEM 5 independent experiments) were sensitive to endoH, further supporting a GA-bypass route for endogenous proteins (*Figure 9C*) (*Hanus et al., 2016*). Interestingly, while the majority of surface GluA1 was sensitive to endoH, endoH did not completely deglycosylate GluA1 (i.e. the mobility of endoH-treated GluA1 did not match that of PNGase-treated GluA1) suggesting surface GluA1 is either partially processed, or that an undefined post-ER modification is occurring. In contrast to AMPA receptors, TARPγ8 in the same biochemical preparation was completely endoH resistant, consistent with our data demonstrating that not all synaptic cargoes follow a GA-independent trafficking route to the cell surface (*Figure 9C* and *Figure 8—figure supplement 4*).

## Discussion

Local trafficking of newly synthesized synaptic receptors, channels and adhesion molecules in neuronal dendrites is thought to play a central role in maintaining and modulating synaptic function. However, it remains unknown how, or even whether, dendritically translated integral membrane proteins are delivered to nearby sites following ER exit and subsequent trafficking through the labyrinthine cellular secretory network. Our findings reveal that synaptic proteins exiting the dendritic ER

undergo spatially restricted trafficking to dendritic ERGICs. Following ERGIC accumulation, cargo next appears in dendritic REs, which are critical for subsequent PM delivery. Intriguingly, we observed that dendritic secretory cargo trafficking to REs and the PM could occur even when the GA was disrupted. Thus, dendritic secretory cargo can undergo direct, local trafficking through a unique secretory network involving ERGICs and REs, but bypassing the GA (*Figure 10*).

## Local entry of GluA1 into the secretory pathway

While synthesis of integral membrane proteins and secreted factors can occur locally in dendrites, it was unclear whether dendritic secretory organelles could spatially coordinate local delivery of these proteins to nearby dendritic sites. This issue is further complicated by the fact that nascent proteins are able to diffuse within the ER (*Cui-Wang et al., 2012*). While dendritic ER can undergo morphological plasticity to spatially restrict lateral diffusion, it was unknown whether proteins are similarly spatially constrained after they exit the ER and traffic through subsequent secretory organelles.

Previous studies using the vesicular stomatitis viral glycoprotein (VSV-G) to investigate the dendritic secretory network report that following ER exit, a significant fraction of dendritic pre-Golgi vesicle carriers are shuttled towards the somatic GA, suggesting that spatial information may be lost as cargo exits the ER (*Horton and Ehlers, 2003*). However, more recent studies report local trapping of a fraction of VSV-G in ERGIC structures following ER exit, supporting the possibility of a local

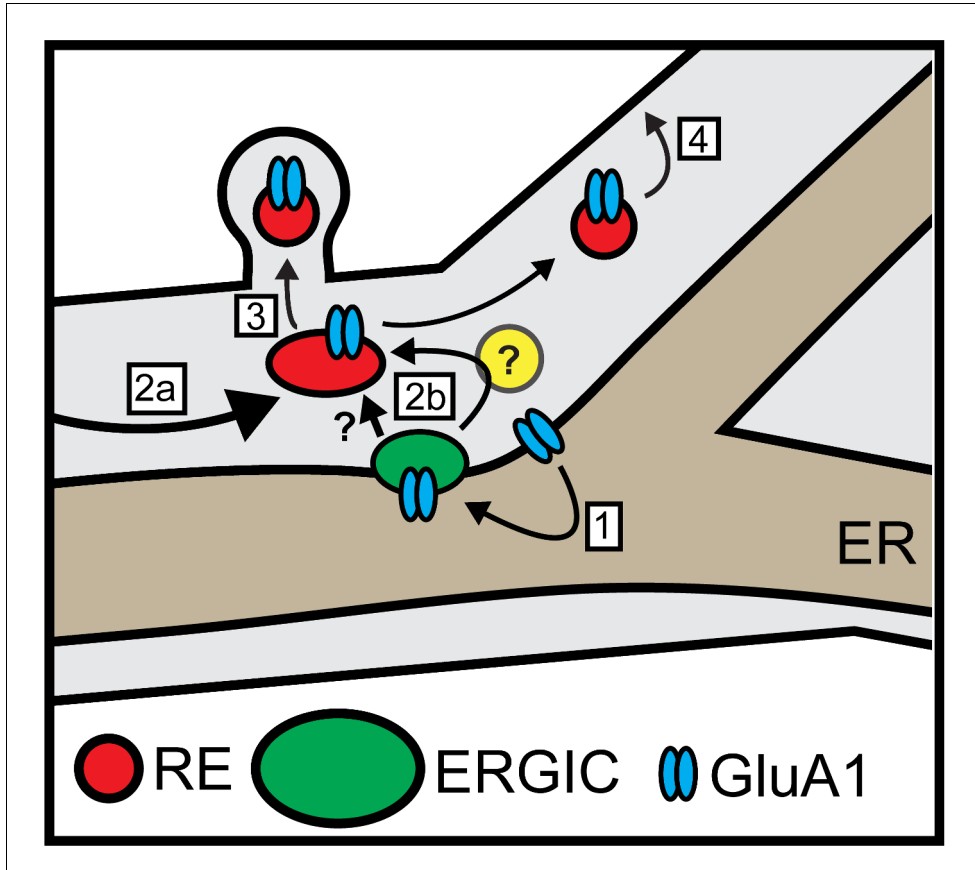

**Figure 10.** Model for local dendritic trafficking. GluA1 exiting the dendritic ER undergoes local entry into nearby ERGIC (step 1). GluA1 subsequently enters the recycling compartment, either through direct ERGIC/RE coupling or through an unidentified intermediate organelle that does not contain conventional GA markers (yellow circle) (step 2b). Cargo can also enter REs through conventional post-GA trafficking that takes place in the soma, followed by long-range transport to dendrites (step 2a). In addition to mobile REs in the dendritic shaft, GluA1 is also directed to spine-resident REs (step 3). The RE compartment mediates a substantial fraction of surface delivery of GluA1 (step 4).
DOI: https://doi.org/10.7554/eLife.27362.035

secretory network (*Chen et al., 2013*; *Hanus et al., 2014*). Importantly, we find that the synaptic proteins GluA1 and NL1 follow the same initial itinerary as VSV-G, revealing a general dendritic trafficking principal that applies to diverse secretory cargoes. However, based on the BFA sensitivity of NL1 and VSV-G surface expression, their trafficking itineraries must diverge from GluA1 following ERGIC accumulation.

We also made the surprising observation that the dendritic and somatic trafficking networks were strikingly segregated. Using a local photoconversion strategy, we found that the majority of GluA1 released from the dendritic ER was effectively retained within nearby dendritic ERGICs. Little GluA1 accumulated in the somatic Golgi, even though the ER is contiguous between these compartments, and mobile vesicles can traverse the dendrite/soma boundary. Conversely, GluA1 released from the somatic ER was efficiently captured in the somatic GA, with little cargo accumulating in dendritic ERGICs. Thus, the downstream organelles and processing of secretory proteins largely depends on their location of ER exit, which would have a major impact on post-translational modifications such as N-linked glycosylation (*Hanus et al., 2016*).

## A role for Recycling Endosomes in dendritic anterograde trafficking

Following ERGIC accumulation, we observed that dendritic cargo trafficked to mobile vesicles positive for the RE markers TfR and Rab11. While the RE network plays a central role in recycling proteins to and from the PM, it has also been reported to mediate anterograde secretory trafficking of a number of different proteins in other cell types (*Ang et al., 2004*; *Lock and Stow, 2005*; *Cancino et al., 2007*; *Thuenauer et al., 2014*). In neurons, REs are widely distributed throughout dendrites, where they play a critical role in integrating synaptic activity to regulate the protein composition of the neuronal plasma membrane (*Park et al., 2004*; *Lasiecka and Winckler, 2011*; *Kennedy and Ehlers, 2011*; *Esteves da Silva et al., 2015*). While synaptic proteins, such as AMPA receptors can undergo basal and activity-induced recycling through the RE network, our findings expand the role of dendritic REs to include anterograde trafficking of newly secreted synaptic cargo molecules. Because REs are mobile, it is unclear whether the spatial distribution of cargo leaving the ER and ERGIC network is preserved once it accesses REs, but previous studies suggest that the dendritic RE network is spatially restricted near clusters of dendritic spines (*Cooney et al., 2002*). Indeed, REs migrate into dendritic spines in an activity-dependent manner where they may regulate synaptic properties (*Esteves da Silva et al., 2015*; *Park et al., 2006*). Our data indicate that REs could maintain and regulate dendritic and synaptic properties through constitutive delivery of newly synthesized integral membrane and secreted proteins, including to a select subset of synapses via spine-resident REs. Given the frequent juxtaposition of spine REs and the spine apparatus, it is tempting to speculate that REs receive locally generated cargo from the ER-derived spine apparatus for targeted synaptic delivery.

## Bypassing the Golgi Apparatus in dendrites

Previous reports indicate that a subset of cargo molecules, including CFTR (*Yoo et al., 2002*), connexins (*Martin et al., 2001*) and the drosophila anti-PS1 integrin (*Schotman et al., 2008*) can traffic to the cell surface without traversing the GA. These cargoes may have intrinsic signals that direct them on a trajectory that bypasses the GA. However, in neurons, the large majority of dendrites lack detectable GA, necessitating a bypass mechanism for a larger set of secreted and integral membrane molecules. To assess whether dendritic ER cargo can bypass the GA, we performed our trafficking assays in the presence of BFA to disrupt GA function. We found that GluA1 released from dendritic ER could still access the local RE network, even in BFA-treated cells where the somatic GA was disrupted. Accordingly, a significant fraction of GluA1 was delivered to the PM in the continued presence of BFA. These data provide compelling evidence for a GA bypass route in neurons, but whether this occurs through direct ERGIC to RE trafficking, or through an intermediate GA-like organelle (that lacks canonical GA markers) remains an open question (*Figure 10*) (*Mikhaylova et al., 2016*).

Similar to classic GA temperature block experiments, trafficking of secretory cargo from dendritic ERGIC to REs and to the PM is completely blocked at 20 °C, (*Matlin and Simons, 1983*), indicating that dendritic ERGICs and the somatic GA share some properties. However, the somatic GA was rapidly and robustly dismantled following BFA treatment, while dendritic ERGICs remained intact.

Whether dendritic ERGICs harbor the same secretory processing enzymes as somatic GA remains an intriguing and open question, but our data support a previous study demonstrating that a diverse cohort of synaptic receptors, channels and cell adhesion proteins present on the neuronal cell surface lack mature N-glycosylation, a hallmark of GA processing (*Hanus et al., 2016*). Thus, local secretory processing in dendrites may yield an exclusive repertoire of posttranslational modifications that could uniquely impact protein structure and function.

## Materials and methods

### Key resources table

| Reagent type (species) or resource | Designation | Source or reference | Identifiers | Additional information |
|---|---|---|---|---|
| Rattus Norvegicus | Sprague Dawley | Charles river | | Both males and females used in this study |
| transfected construct (3xFM-FP-GluA1) | 3xFM-GluA1 | this paper | | Generated from GluA1 cDNA, gift from Dr. Michael Ehlers |
| transfected construct (4xFM-FP-Nlgn1) | 4xFM-NL1 | this paper | | Generated from neuroligin 1 construct obtained from Dr. Peter Scheiffele's lab, addgene clone #15262 |
| transfected construct (VSV-G-4xFM-YFP) | VSV-G-YFP-4xFM | this paper | | Generated from VSV-G cDNA from Dr. Michael Ehlers |
| transfected construct (Rab11/Rab11DN) | GFP-Rab11 | | Derived from Addgene (12674) | Gift from Dr. Richard Pagano |
| transfected construct (Rab8/Rab8DN) | GFP-Rab8 | | Derived from Addgene (24898, 24899) | Gift from Dr. Maxence Nachury |
| transfected construct (Rab5/Rab5DN) | GFP-Rab5 | | Derived from Addgene (28045) | Gift from Dr. Qing Zhong |
| transfected construct (ERGIC53) | GFP-ERGIC53 | PMID: 20639694 | Addgene (38270) | Gift from Dr. Noboru Mizushima |
| transfected construct (TfR) | TfR-mCh/GFP | PMID: 20434989 | | Gift from Dr. Michael Ehlers |
| transfected construct (4xFM-mCh) | 4xFM-mCh | this paper | | Generated from mCh N1 vector (Clontech) and Ariad regulated secretion kit 4xFM plasmid. |
| transfected construct (GalT) | | | Addgene (54108) | Gift from Dr. Michael Davidson |
| transfected construct (TfR-mCh-KDEL) | | this paper | | Generated from TfR cDNA obtained from Dr. Michael Ehlers |
| transfected construct (GFP-synaptopodin) | GFP-Synpo | PMID: 25164660 | | Gift from Dr. Menahem Segal |
| antibody | HA | Biolegends | clone 16B12; #901501; RRID:AB_2565006 | 1:1000 |
| antibody | BIP | abcam | ab21685; RRID:AB_2119834 | 1:1000 |
| antibody | GM130 | BD | 610822; RRID:AB_398141 | 1:1000 |
| antibody | VSVG | Kerafast | 8G5F11 | 1:1000 |
| antibody | TGN38 | Gift from Kathryn Howell | 2F7 | 1:1000 |
| antibody | mCh | abcam | ab167453; RRID:AB_2571870 | 1:1000 IF; 1:5000 for western |
| antibody | GFP | Neuromab | N86/6 RRID:AB_2313651 | 1:1000 |
| antibody | GFP | Invitrogen | A11122; RRID:AB_221569 | 1:1000 |
| antibody | ERGIC53/p58 | Sigma | E1031; RRID:AB_532237 | 1:500 |

*Continued on next page*

*Continued*

| Reagent type (species) or resource | Designation | Source or reference | Identifiers | Additional information |
|---|---|---|---|---|
| peptide, recombinant protein | Thrombin | Sigma | T6884 | 1 U/ml |
| chemical compound, drug | Brefeldin A | Tocris | Product #1231 | 5 µg/ml |
| chemical compound, drug | DD Solubilizer | Clontech | Product #635054 | 1 µM |

## Molecular cloning

All expression constructs used in this study are listed in the table below. All constructs were verified by sequencing at the Barbara Davis Sequencing Core at the UCSOM. $3xF_M$-GluA1 constructs were made by inserting $3xF_M$ repeats and a fluorescent tag bearing a thrombin site and hemagglutinin tag into a vector containing the GluA1 ORF (GluA1 backbone was a gift from Michael Ehlers, Biogen). The entire cassette was subcloned into a custom CAG promoter backbone to generate the constructs used in the study. Different fluorescent tags were PCR amplified and subcloned into this construct. $4xF_M$-NL1 was generated by insertion of NL1 ORF (Addgene #15260, a gift from Peter Schieffelle) into a construct containing $4xF_M$. This fusion was then subcloned into a pCAG backbone. The SNAPtag domain was amplified from addgene plasmid #29652 (a gift from Eric Cambeau). To generate the dominant-negative Rab11 mutant, eGFP-Rab11(WT) was subjected to site-directed mutagenesis and mCherry was subcloned in place of the eGFP tag. Wildtype Rab5 was generated by subcloning the Rab5 ORF from addgene plasmid #28043 into Rab5[S34N]. $4xF_M$-mCh was produced by using restriction digest and blunt re-ligation to introduce a stop codon after AA 4 of the NL1 ORF in the $4xF_M$-mCh-NL1. TfR-Halotag was made by PCR amplifying the HaloTag domain from addgene #29644 (a gift from Eric Cambeau) and generating a C-terminal fusion with TfR.

## Constructs used in this study

| Construct | Fusion | Tag(s) | Epitope | Variants | Promoter | Signal peptide | Source | Gift from |
|---|---|---|---|---|---|---|---|---|
| $3xF_M$-FP-GluA1 | N | mEOS3.2, mCh, SEP, SNAP, dFP | HA | - | pCAG | Endogenous | | |
| $4xF_M$-FP-Nlgn1 | N | mCh, SEP, SNAP | HA | - | pCAG | Growth hormone | | |
| VSV-G-$4xF_M$-FP | C | YFP | - | | pCMV | | | |
| Rab11 | N | mCh, GFP | HA | [S25N] | pCMV | - | Addgene (12674) | Dr. Richard Pagano |
| Rab8 | N | GFP | - | [T22N] | pCMV | | Addgene (24898, 24899) | Dr. Maxence Nachury |
| Rab5 | N | GFP | - | [S34N] | pCMV | - | Addgene (28045) | Dr. Qing Zhong |
| ERGIC53 | N | GFP | - | - | pCMV | Calreticulin | Addgene (38270) | Dr. Noboru Mizushima |
| TfR | C | GFP, mCh, HALO | - | - | pCMV | - | | |
| $4xF_M$-mCh | N | mCh | - | - | pCAG | - | | |
| GalT | C | mEmerald | - | - | pCMV | Endogenous | Addgene (54108) | Dr. Michael Davidson |
| TfR-KDEL | C | mCh | - | - | pCMV | - | | |
| Cell Fill | - | mCh, GFP | - | - | pCMV and pCAG | - | | |
| GFP-synaptopodin | N | GFP | - | - | pCMV | - | | Dr. Menahem Segal |

## Cell culture

Primary cortical neurons were prepared from neonatal Sprague Dawley rats of either sex. Frontal cortex was separated from the brains of postnatal day 0–2 rats and dissociated by papain digestion. Neurons were plated at 200,000 cells/well in MEM and 10% FBS (Hyclone) containing penicillin/streptomycin on poly-D-lysine-coated 18 mm glass coverslips placed in a 12 well dish. After 1 d, the medium was replaced with Neurobasal-A media (NBA) supplemented with B27 (Invitrogen) and GlutaMAX (Thermo Fisher; Waltham, MA). The neurons were then fed with NBA, B27, and mitotic inhibitors (uridine fluoro deoxyuridine) by replacing half of the medium on day 6 or day 7 and then once

weekly. Neurons were maintained at 37°C in a humidified incubator at 5% CO2. Neurons were transfected using Lipofectamine 2000 (Invitrogen) according to the manufacturer's recommendations and allowed to express for 48 hr. All neurons were between 15 days in vitro (DIV15) and DIV18 at the time of the experiment. For HeLa and COS7 cultures, cells were maintained in DMEM supplemented with 10% FBS (Hyclone) and penicillin/streptomycin. Cells were passaged approximately every 2 days by trypsinization and resuspension in fresh DMEM. Heterologous cells were transfected using Lipofectamine 2000 (Invitrogen) according to the manufacturer's recommendations and allowed to express for 24 hr. COS7 and HeLa cell lines were obtained from ATCC, expanded and frozen. Parent cell lines were freshly thawed and validated by cellular morphology and growth characteristics. Recent (May, 2017) mycoplasma tests were negative.

## Inducible-release trafficking

DD solubilizer (1 μM final concentration; Clontech cat # 635054) was added to live neurons in NBA + B27 and incubated for the indicated period of time before surface labeling, fixation or live-cell imaging. For temperature block experiments, cells were transferred to a chamber maintained at 20°C for 2 or 4 hr as indicated. For thrombin cleavage, thrombin was added to live cells at 1 U/mL (sigma T6884). For Golgi-bypass experiments, cells were treated with BFA at 5 μg/mL for the final 20 min of temperature block before transfer to 37°C for 1 hr (for live-cell microscopy) or 2 hr (surface-labelling). For all experiments we confined our analysis to neurons with a pyramidal-shaped cell body, large apical dendrite and presence of dendritic spines. While these morphological criteria will bias our analyses to excitatory neurons it is possible that some inhibitory neurons were analyzed.

## Live-cell surface labelling and immunocytochemistry

Live cell surface labeling was performed for cells expressing $3xF_M$-GluA1 or $4xF_M$-Nlgn1 using anti-HA (Biolegends, clone 16B12 #901501 RRID:AB_2565006; 1:1000) or for experiments using $4xF_M$-VSV-G using anti-VSV-G (Kerafast, clone 8G5F11, 1:1000). Cells were incubated in NBA + B27 with antibody at 37°C for 15 min. Cells were washed once with artificial cerebrospinal fluid (ACSF) solution containing the following (in mM): 130 NaCl, 5 KCl, 10 HEPES, 30 glucose, 2 CaCl$_2$, 1 MgCl$_2$,. 002 TTX, pH 7.4. Cells were then fixed with 4% PFA for 10 min at room-temperature and labelled with fluorescent-conjugated secondary for 30 min in non-permeabilizing conditions. For immunocytochemistry, cells were fixed in 4% PFA, permeabilized with 0.1% Triton X-100 or 0.5% Tween-20 for 10 min and blocked for 30 min at room temperature with 5% BSA. Cells were incubated with primary antibodies at the reported dilutions for 60 min at room temperature, washed in PBS, incubated with fluorescent-conjugated secondary antibodies for 60 min at room temperature and mounted (Prolong Gold, Life technologies). Primary antibodies used in this study include BIP (abcam; ab21685 RRID AB_2119834; 1:1000), GM130 (BD; 610822 RRID AB_398141; 1:1000), TGN38 (clone 2F7 was a gift from Kathryn Howell), mCh (abcam; 167453 RRID AB_2571870; 1:1000), GFP (Invitrogen; A11122 RRID AB_221569; 1:1000 or neuromab; clone N86/6 RRID AB_2313651; 1:1000) and ERGIC/p58 (Sigma; E1031 RRID AB_532237, 1:500). Fluorescent-conjugated secondary antibodies include goat anti-mouse alexa fluor 647 and goat anti-rabbit alexa fluor 647 or alexa fluor 568 (Life technologies, 1:1000)

## Surface biotinylation and deglycosylation

Surface biotinylation was carried out essentially as in *Hanus et al. (2016)*. Dissociated neurons were washed in imaging buffer containing (in mM): 120 NaCl, 3 KCl, 2 CaCl$_2$, 2 MgCl$_2$, 15 glucose, 10 HEPES pH 7.4. Cells were then treated with the same buffer containing NHS-SS-biotin (0.8 to 1 mg/mL, Thermo) at room temperature for 7 min. Cells were rinsed and excess biotinylating reagent was quenched using the same buffer supplemented with 10–20 mM L-lysine. Cells were lysed in PBS containing 1% triton X-100, 0.6%SDS and a protease inhibitor cocktail. Biotinylated proteins were purified from cell lysates over streptavidin-conjugated agarose beads and eluted by reduction of disulfide-linked biotin with 50 mM DTT for 15 min at 75 °C. Purified surface fractions were divided and either left untreated or treated with endoHf (New England Biolabs) or PNGase (New England Biolabs) according to the manufacturer's insctructions. Extracts were diluted (~1.5 fold) in sodium phosphate (50 mM, pH 5.5 final) or sodium citrate buffer (50 mM pH 7.5) plus NP40 (or triton X-100,

1% final) for PNGase and endoH respectively. Enzymes were used at 1000 (PNGase) or 3000 (endoH) units/µg total protein and incubated overnight at 37 ˚C.

## Confocal microscopy

Live-cortical neurons were imaged in ACSF at 34 ˚C on an Olympus IX71 equipped with a spinning disc scan head (Yokogawa) with a 60x NA1.4 objective. Excitation illumination was delivered from an AOTF controlled laser launch (Andor) and images were collected on a 1024 × 1024 pixel Andor iXon EM-CCD camera. Data acquisition was performed with Metamorph (Molecular Devices) or Andor IQ software. For some experiments a spinning disk Marianas live-cell imaging system (3i) running Slide-book software (3i) was used. Photoconversion of mEOS3.2 expressing cells was carried out using targeted illumination (FRAPPA system, Andor) with the 405 nm laser (1 s dwell time 5% laser power). For most experiments, a 7 µm Z-stack (0.5 µm step-size) was acquired at each time point.

## Vesicle tracking assay

At the indicated time after ER-release, neurons were placed in ACSF and imaged. Segments of dendritic arbor were bleached with the 488 nm laser using a FRAPPA targeted illumination unit (Andor). After bleaching, single-plane imaging was conducted at either four hz (dual bandpass emission filter) or three hz (switching between GFP and RFP emission filter sets). Controls were performed to ensure cross-excitation did not contaminate either the green or red channel when using the dual bandpass emission filter.

## SNAP/Halotag labelling

HaloTag or SNAPTag ligand-dye conjugates were added to the cell culture media at a concentration of 100 nM. Cells were incubated for 20 min, washed twice with fresh NBA + B27. Cells were then incubated in a 50:50 mixture of fresh and conditioned NBA + B27 for at least 30 min before imaging or fixation.

## STED microscopy

Neurons were labelled with a STED-compatible secondary antibody pair (STAR580/STAR635p; Abberior; 1:1000) and imaged on a custom built STED microscope (*Meyer et al., 2016*). Resolution was estimated using the equation (bead size = 40 nm):

$$resolution = \sqrt{(FWHM)^2 + (beadsize)^2}$$

## Biochemistry

COS7 cell supernatant and pellets were harvested at the indicated time points after addition of DDS. Samples were boiled in laemelli sample buffer (Biorad) and run on a 10% polyacrylamide gel and transferred to PVDF membrane. Blots were incubated overnight at 4°C with an antibody to mCh (abcam; 167453;1:5000) and detected with HRP-conjugated anti-rabbit secondary (Biorad; 1706515, 1:5000). Blots were developed using Supersignal West Dura Extended Duration Substrate (Thermo; 34075).

## Data analysis

All image analysis was performed on raw images in ImageJ (*Schindelin et al., 2012*). Generation of graphs and statistical analysis were carried out in Prism 7 (Graphpad). For all analysis, maximum intensity projections of each acquired Z-stack were used. For quantification of surface-labelling intensity after induction of ER-release, images were analyzed by automatically generating a mask based on neuronal morphology, determining the fluorescence intensity in each channel under the mask, and subtracting the background intensity based on a user-defined background region. For randomization of TfR puncta, a Matlab (Mathworks) code was written that randomly redistributes an equal number of puncta within the boundaries of the neuronal dendrite. For analysis of vesicular co-trafficking, a single time-point with low vesicle number in each fluorescent channel was selected. Coordinates of each vesicle were determined using the find maxima function in ImageJ. The distance between maxima coordinates was determined using the JACOP plugin for ImageJ (*Bolte and Cordelières, 2006*) and distances of less than 500 nm were considered to be colocalized. For each

identified vesicle, cotrafficking was manually verified in the original time-lapse series. For ERGIC/RE size determination based on STED imaging the Squassh plugin for ImageJ was used (*Rizk et al., 2014*).

### Statistical analysis

Unless otherwise indicated, all data is displayed as mean value ± standard error of the mean. For pairwise comparison of normally distributed data sets, unpaired 2-tailed Student's t-test was used. For comparison of cumulative distributions, the Kolmogorov–Smirnov test was used.

## Acknowledgements

We would like to thank Drs. Mark Dell'Acqua and Chandra Tucker for critical review and discussions of this manuscript. We would like to thank Dr. Luke Lavis (Janelia Farm Research Campus) for generously providing JF646 Halotag ligand. We thank Emily Gibson, Annie Mauborgne, Dorian Miremont and Aurélie Jervaise for excellent technical assistance. We also thank and Stephanie Meyer, Radu Moldovan and Dominik Stich of the University of Colorado Advanced Light Microscopy Core. The STED microscope was funded through an NSF Major Research Instrumentation grant #DBI-1337573 and an NIH shared Instrument Grant #S10 RR023381. This work was supported by NS092421 (ABB); National Science Foundation DGE-1553798 and a Howard Hughes Medical Institute Gilliam Fellowship (AMB); Agence Nationale de la Recherche (ANR-16-CE16-0009-01), Paris-Descartes University and INSERM (CH); NS082271, The Pew Charitable Trusts and The Boettcher Foundation (MJK).

## Additional information

### Funding

| Funder | Grant reference number | Author |
| --- | --- | --- |
| National Institute of Neurological Disorders and Stroke | RO1 NS082271 | Matthew J Kennedy |
| McKnight Endowment Fund for Neuroscience | | Matthew J Kennedy |
| Pew Charitable Trusts | | Matthew J Kennedy |
| National Institute of Neurological Disorders and Stroke | F30 NS092421 | Aaron B Bowen |
| Howard Hughes Medical Institute | Gilliam Fellowship | Ashley M Bourke |
| Institut National de la Santé et de la Recherche Médicale | | Cyril Hanus |
| National Science Foundation | DGE-1553798 | Ashley M Bourke |
| Agence Nationale de la Recherche | ANR-16-CE16-0009-01 | Cyril Hanus |

The funders had no role in study design, data collection and interpretation, or the decision to submit the work for publication.

### Author contributions

Aaron B Bowen, Conceptualization, Formal analysis, Funding acquisition, Investigation, Writing—original draft, Writing—review and editing; Ashley M Bourke, Cyril Hanus, Data curation, Formal analysis, Funding acquisition, Investigation, Methodology, Writing—review and editing; Brian G Hiester, Data curation, Formal analysis, Investigation, Writing—review and editing; Matthew J Kennedy, Conceptualization, Resources, Supervision, Funding acquisition, Methodology, Writing—original draft, Writing—review and editing

## Author ORCIDs
Aaron B Bowen (iD) http://orcid.org/0000-0002-4465-8627
Matthew J Kennedy (iD) http://orcid.org/0000-0002-7029-7802

## Ethics

Animal experimentation: This study was performed in strict accordance with the recommendations in the Guide for the Care and Use of Laboratory Animals of the National Institutes of Health. All animal procedures were carried out in accordance with a protocol approved by the University of Colorado Denver Institutional Animal Care and Use Committee (protocol # B-98715(04)1E).

## Decision letter and Author response

Decision letter https://doi.org/10.7554/eLife.27362.039
Author response https://doi.org/10.7554/eLife.27362.040

---

## Additional files

### Supplementary files

• Transparent reporting form
DOI: https://doi.org/10.7554/eLife.27362.038

---

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
