## [Decision Letter]

Thank you for submitting your article "Golgi-independent secretory trafficking through recycling endosomes in neuronal dendrites and spines" for consideration by *eLife*. Your article has been reviewed by three peer reviewers, and the evaluation has been overseen by a Reviewing Editor and Anna Akhmanova as the Senior Editor. The following individuals involved in review of your submission have agreed to reveal their identity: Jason D Shepherd (Reviewer #2).

The reviewers have discussed the reviews with one another and the Reviewing Editor has drafted this decision to help you prepare a revised submission.

Summary:

The manuscript by Bowen and Kennedy describes a series of elegant experiments designed to track the movement of membrane proteins from ER to synaptic membranes. While the study of this type of intracellular trafficking is not new territory, the authors approach the question from a novel perspective by focusing on regions of dendrites lacking a traditional Golgi assembly (GA) that would normally be involved in such a pathway. The evidence provided by Bowen and Kennedy, particularly using their sophisticated inducible ER-release technique, indicate an involvement of recycling endosomes in membrane protein trafficking to the plasma membrane. Overall, the study is well-executed and the findings are of significant importance to the fields of neuronal protein trafficking. However, there remain two primary concerns, shared by the three reviewers of this work, that should be addressed.

Major Concerns – Consensus of all three reviewers:

1) Given that the paper relies heavily on creating an artificial aggregate of GLUA1 in the ER there is a potential worry that the trafficking route studied (ER-ERGIC-RE-Surface) is one that is not normally populated by GLUA1. Along this same line of logic, the emphasis on protein over-expression in cultured neurons could also create a bias toward the trafficking route emphasized in the current data. If the authors could come up with an experiment that gives some quantitative information about the population of GLUA1 the (ER-ERGIC-RE-Surface) pathway under normal conditions (ideally endogenous receptors), it would greatly strengthen the conclusiveness of the work. One possibility would be to determine roughly what fraction of protein normally goes through one route or the other. One can imagine trying to design a strategy where one looks at how much of a protein that makes it to a cell surface (without pre-aggregation in the ER) gets exposed to the enzymes of the GA, such as furin. Without such an experiment there may be more room to doubt the significance of the central findings of this manuscript. This is not a prescriptive experimental suggestion. The authors almost certainly will have ideas to address this general concern that they would consider feasible within the time-frame established by the *eLife* review process.

2) One incongruity in the experiments is the fact that is takes ~ 12 hours to reach steady-state surface concentrations of DDS release Glua1 yet many of the experiments looking at the confinement of the trafficking to ERGIC or RE are only carried out over ~2 or at most 4 hour time periods. When you photo convert the soma and release Glua1 with DDS and wait 12 hours where does this protein end up? It is possible that on these time scales it still ends up going out to dendrites and spines, which would be contrary to the principal conclusions and show that both routes are used equally. This should be straightforward to address.

---

## [Author Response]

[…] 1) Given that the paper relies heavily on creating an artificial aggregate of GLUA1 in the ER there is a potential worry that the trafficking route studied (ER-ERGIC-RE-Surface) is one that is not normally populated by GLUA1. Along this same line of logic, the emphasis on protein over-expression in cultured neurons could also create a bias toward the trafficking route emphasized in the current data. If the authors could come up with an experiment that gives some quantitative information about the population of GLUA1 the (ER-ERGIC-RE-Surface) pathway under normal conditions (ideally endogenous receptors), it would greatly strengthen the conclusiveness of the work. One possibility would be to determine roughly what fraction of protein normally goes through one route or the other. One can imagine trying to design a strategy where one looks at how much of a protein that makes it to a cell surface (without pre-aggregation in the ER) gets exposed to the enzymes of the GA, such as furin. Without such an experiment there may be more room to doubt the significance of the central findings of this manuscript. This is not a prescriptive experimental suggestion. The authors almost certainly will have ideas to address this general concern that they would consider feasible within the time-frame established by the eLife review process.

We attempted the experiment suggested by the reviewers by assaying whether a fraction of GluA1 makes it to the plasma membrane without encountering furin protease (which is present in the Golgi network). However, numerous groups have reported that furin is not solely localized to the Golgi network but is also present in the plasma membrane and the endosomal network (Liu et al., JCB 139:1719; Simmen et al., Mol. Cell. Biol. 19:3136; Teuchert et al., JBC 274:36781). Accordingly, we did not detect any surface protein that had not been processed by furin (our approach was to surface label neurons expressing GluA1 harboring an epitope that is removed by furin and then surface label cells with antibodies against the epitope). However, the broad localization of furin at the plasma membrane and within endosomes makes this experiment difficult to interpret.

As an alternate approach to address this issue, we collaborated with Dr. Cyril Hanus (INSERM, Paris) who recently demonstrated that a large fraction of neuronal surface proteins display immature N-linked glycosylation (i.e. sensitive to the deglycosylating enzyme endoH), consistent with a GA bypass trafficking route (*eLife* 5:e20609). We have generated a new data set quantifying the fraction of endogenous surface AMPA receptors that display immature (high mannose) N-linked glycosylation assayed by endoH sensitivity. We found that 40 ± 3% of surface GluA2 and 88 ± 1% of surface GluA1 display endoH sensitivity, further supporting a GAbypass pathway for endogenous proteins (discussed in subsection “Golgi-independent transfer of GluA1 to the PM through ERGIC and Res”). We included these data in Figure 9 of the revised manuscript.

2) One incongruity in the experiments is the fact that is takes ~ 12 hours to reach steady-state surface concentrations of DDS release Glua1 yet many of the experiments looking at the confinement of the trafficking to ERGIC or RE are only carried out over ~2 or at most 4 hour time periods. When you photo convert the soma and release Glua1 with DDS and wait 12 hours where does this protein end up? It is possible that on these time scales it still ends up going out to dendrites and spines, which would be contrary to the principal conclusions and show that both routes are used equally. This should be straightforward to address.

We defined the trafficking itinerary of GluA1 at early times (2-4 hours following ER release) to ensure our analysis was focused on cargo making its initial voyage to the PM. This timeframe corresponds to when GluA1 begins appearing on the cell surface (Figure 1). Numerous other factors (endocytosis, recycling, degradation etc.) are likely to contribute to the long-term steady state surface level at later times. Nevertheless, this point deserves further clarification. Our experiments do not support a strict segregation of cargo between the somatic and dendritic compartments once the cargo progresses beyond the GA. For example, we observed a reduction in the fraction of dendritic REs that were positive for newly released GluA1 when we treated cells with brefeldin A (Figure 8 of the revised manuscript). This result suggests that under normal conditions a significant fraction of RE-localized cargo in dendrites originated from the somatic GA (included in the model in Figure 10 of the revised manuscript).

Indeed, direct GA to RE trafficking has been previously described in heterologous cells (Ang et al., J. Cell Biol. 167:531) and is further supported by our data. To more directly demonstrate that somatic GA produces mobile GluA1 carriers that can enter dendrites, we performed a new experiment where we conditionally released mCh-GluA1 from the ER and allowed it to accumulate in the somatic GA and dendritic ERGICs. We then photobleached all detectable signal in the dendritic arbor (preserving the somatic GA signal) and then performed rapid timelapse imaging 1 hour later. After the recovery period, we observed both mobile and stationary dendritic puncta that decreased in intensity and frequency as a function of distance from the soma. Since we had completely eliminated dendritic signal beforehand, this signal must have originated from the somatic GA. Thus, in addition to local trafficking routes, long-range transport from the somatic GA can also occur, especially in regions proximal to the soma. Whether these two pathways remain segregated or merge prior to PM delivery will be a topic of future experiments. We have now included these new data in Figure 4—figure supplement 4 of the revised manuscript and included a more thorough discussion in the Results section (subsection “Synaptic cargo accumulates in dendritic ERGIC following ER exit”) to clarify this point.